

# Sources and processes sustaining surface $CO_2$ and $CH_4$ fluxes in a tropical reservoir: the importance of water column metabolism

Cynthia Soued[1], Yves T. Prairie[1]

[1]Groupe de Recherche Interuniversitaire en Limnologie et en Environnement Aquatique (GRIL), Département des Sciences
Biologiques, Université du Québec à Montréal, Montréal, H2X 3X8, Canada.

*Correspondence to*: Cynthia Soued (cynthia.soued@gmail.com)

**Abstract.** Freshwaters are important emitters of carbon dioxide ($CO_2$) and methane ($CH_4$) to the atmosphere, two potent
greenhouse gases (GHG). While aquatic surface GHG fluxes have been extensively measured, there is much less information
about their underlying sources. In lakes and reservoirs, surface GHG can originate from horizontal riverine flow, the
hypolimnion, littoral sediments, and water column metabolism. These processes are generally studied separately, leading to a
fragmented assessment of their relative role in sustaining $CO_2$ and $CH_4$ surface fluxes. In this study, we quantified sources /
sinks of $CO_2$ and $CH_4$ in the epilimnion along a hydrological continuum in a permanently stratified tropical reservoir (Borneo
Island). Results showed that horizontal inputs are an important source of both $CO_2$ and $CH_4$ (18 to 100 % of surface emissions)
in the upstream reservoir branches. However, this contribution fades along the hydrological continuum, becoming negligible
in the main basin of the reservoir, where $CO_2$ and $CH_4$ are uncoupled and driven by different processes. In the main basin,
vertical $CO_2$ inputs and sediment $CH_4$ inputs contributed to on average 60 and 23 % respectively to the surface fluxes of the
corresponding gas. Water column metabolism exhibited wide amplitude and range for both gases, making it the most influential
but uncertain component in the epilimnetic gas budgets. Overall our results show that while sources sustaining surface $CO_2$
and $CH_4$ fluxes vary spatially and between the two gases, internal water metabolism remains a dominant driver. However, this
study also highlights challenges and knowledge gaps related to estimating ecosystem-scale $CO_2$ and $CH_4$ metabolism, which
hinder aquatic GHG flux predictions.

## 1 Introduction

Surface inland waters are globally significant sources of greenhouse gases (GHG) to the atmosphere, namely carbon dioxide
($CO_2$) and methane ($CH_4$) (DelSontro et al., 2018a). Freshwaters act as both transport vessels for terrestrial carbon (C) and as
active biogeochemical processors, making them key sites of GHG exchange with the atmosphere (Tranvik et al., 2018). The
impoundment of rivers for hydropower generation, irrigation, flood control or other purposes, changes the landscape and its C
cycling (Maavara et al., 2017), often resulting in increased aquatic $CO_2$ and $CH_4$ emissions due to the decay of flooded organic
matter (Prairie et al., 2018; Venkiteswaran et al., 2013). Globally, reservoirs are estimated to emit between 0.5 and 2.3 $PgCO_2eq$



yr$^{-1}$ (Barros et al., 2011; Bastviken et al., 2011; Deemer et al., 2016; St. Louis et al., 2000), and this number is predicted to increase with a rapid growth of the hydroelectric sector in the upcoming decades (Zarfl et al., 2015). Several studies have focused on quantifying GHG surface diffusion from reservoirs around the world and have found extremely high variability temporally and spatially (Barros et al., 2011; Deemer et al., 2016), as is found in natural lakes (DelSontro et al., 2018a; Raymond et al., 2013). However, less research exists on the relative contribution of the different sources and processes

sustaining surface diffusive fluxes and their variability, especially in reservoirs.

GHG sources to surface waters can be both internal and external. The magnitude of allochthonous inputs, namely terrestrial organic and inorganic C, is known to increase with soil-water connectivity (Hotchkiss et al., 2015), and with soil C content and leaching capacity (Kindler et al., 2011; Li et al., 2017; Monteith et al., 2007). Soil-derived gas inputs are also temporally variable, generally increasing with discharge, like during storm events (Vachon and del Giorgio, 2014) or rainy seasons (Kim

et al., 2000; Zhang et al., 2019). Terrestrial inputs in the form of organic C can indirectly sustain surface GHG emissions by fueling lake / reservoir *in situ* organic matter respiration (Karlsson et al., 2007; Pace and Prairie, 2005; Rasilo et al., 2017).

The net internal balance between production and consumption processes of $CO_2$ and $CH_4$ influence their surface fluxes. For $CO_2$, aerobic ecosystem respiration (ER) and gross primary production (GPP) are highly variable in space and time, and generally a function of temperature, organic C content and nutrients (Hanson et al., 2003; Pace and Prairie, 2005; Prairie et al.,

1989; Solomon et al., 2013). Net heterotrophy (ER > GPP) is mainly associated with systems receiving high external inputs of organic C (Bogard et al., 2020; Tank et al., 2010; Wilkinson et al., 2016), while net autotrophy (ER < GPP) has been associated to highly productive nutrient-rich systems (Hanson et al., 2003; Sand-Jensen and Staehr, 2009). However, a large part of the variability in measured metabolic rates remains unexplained (Bogard et al., 2020; Coloso et al., 2011; Solomon et al., 2013) and our ability to predict their net balance is still weak in part because the two processes are often studied separately.

Additionally, anaerobic C transformation adds another level of complexity to the C metabolic balance by decoupling GPP and ER (Bogard and del Giorgio, 2016; Martinsen et al., 2020; Vachon et al., 2020). For instance, acetoclastic methanogenesis can transform organic C to $CH_4$ instead of $CO_2$, and hydrogenotrophic methanogenesis converts $CO_2$ to $CH_4$ without producing $O_2$.

$CH_4$ is known to be mostly produced in both profundal and littoral sediments, and reach the water surface by vertical or lateral

diffusive processes (Bastviken et al., 2008; DelSontro et al., 2018b; Encinas Fernández et al., 2014; Guérin et al., 2016). However, there is increasing evidence that $CH_4$ production in the oxic water column contributes significantly to lakes $CH_4$ emissions (Bižić et al., 2019; Bogard et al., 2014; DelSontro et al., 2018b; Donis et al., 2017; Tang et al., 2014). Methanogenesis can be counter-balanced by the oxidation of $CH_4$ to $CO_2$ mainly in oxic and hypoxic environments (Conrad, 2009; Reis et al., 2020; Thottathil et al., 2019). While several studies have measured rates of $CH_4$ production and oxidation in

lakes and reservoirs, few have quantified the net balance of these two processes at an ecosystem-scale (Bastviken et al., 2008; Schmid et al., 2007; Vachon et al., 2019). According to Vachon et al. (2019), this balance is tightly controlled by physical processes within the water column.



Physical mixing in lakes and reservoirs indirectly impacts C metabolic processes by shaping the $O_2$ profile, and directly affects GHG surface diffusion by controlling the transport of $CO_2$ and $CH_4$ from deep to surface water layers (Barrette and Laprise,

2005; Kreling et al., 2014; Pu et al., 2020). Despite its potential importance (Kankaala et al., 2013), very few studies quantified vertical gas transport and the role of this process in fueling surface GHG emissions. The movement of gases within a system depends on the structure of the water column, which changes spatially along the aquatic continuum. Reservoirs in particular exhibit strong gradients in morphometry and hydrology, translating into high spatial heterogeneity in surface GHG fluxes to the atmosphere (Paranaíba et al., 2018; Teodoru et al., 2011).

Understanding what regulates surface $CO_2$ and $CH_4$ concentrations and fluxes to the atmosphere requires knowledge of the interplay between all physical and biogeochemical processes involved, and how they vary spatially. While a number of studies have assessed some processes individually or by difference, very few have measured all relevant components of the epilimnetic mass-balance simultaneously. Here we report on a field study in a tropical East Asian hydropower reservoir quantifying external inputs, sediments inputs, net $CO_2$ and $CH_4$ metabolism, vertical diffusion from deeper layers and gas exchange at the

air-water interface. This allowed us to estimate the relative contribution of each process in shaping surface GHG emissions from the reservoir, and to test whether the epilimnetic mass balance can be closed. The two major rivers feeding the reservoir flow into two elongated branches, acting as a transition zones, before reaching the main basin. This configuration, common in reservoirs, allowed us to quantify and compare epilimnetic $CO_2$ and $CH_4$ regulation in two morphometrically different areas (reservoir branches and main basin). Overall, the aim of this study is to provide an ecosystem-scale portrait of the processes

sustaining surface $CO_2$ and $CH_4$ emissions and examine how they change when transitioning from a river delta to an open basin.

## 2 Materials and methods

### 2.1 Site and sampling description

The study was conducted in Batang Ai hydroelectric reservoir in Sarawak Malaysia (latitude 1.16° and longitude 111.9°). The

reservoir is located on the Borneo Island in a tropical equatorial climate with a constantly high temperature averaging 23 °C and 32 °C during nighttime and daytime respectively (Sarawak Government, 2019). The region experiences two weak monsoon seasons (November to February and June to October) with a yearly average rainfall of 3300 to 4600 mm (Sarawak Government, 2019). The reservoir was impounded in 1985 with a dam wall of 85 m, a surface area of ~68.4 $km^2$ and a watershed area of 1149 $km^2$ of mostly undisturbed forested land (limited rural habitations and small scale croplands).

We distinguish between three sections of the study site: inflows, reservoir branches, and reservoir main basin shown in Figure 1. The inflows are the two main reservoir inlets: Batang Ai and Engkari rivers (3 to 10 m deep where sampled). The two rivers flow into two arms that we refer to as the reservoir branches (10.8 $km^2$, mean and max depths of 18 and 52 m respectively). The reservoir branches merge into the main basin of the reservoir (58.9 $km^2$, mean and max depths of 30 and 73 m respectively). Surface sampling was performed in 36 sites across the three study sections, and water column profile sampling (from 0 up to



32 m, each 0.5 to 3 m) was done in 9 sites in the reservoir branches and main basin (Figure 1). Sampling was repeated (with a few exceptions) during four campaigns: 1) November 14th to December 5th 2016 (Nov-Dec 2016), 2) April 19th to May 3th 2017 (Apr-May 2017), 3) February 28th to March 13th 2018 (Feb-Mar 2018), and 4) August 12th to 29th 2018 (Aug 2018).

## 2.2 Physical and chemical analyses

Water temperature, dissolved oxygen, and pH were measured using a multi-parameter probe (YSI model 600XLM-M) equipped with a depth gauge and attached to a 12 Volt submersible pump (Proactive Environmental Products model Tornado) for water samples collection. Concentrations of dissolved organic carbon (DOC), total phosphorus (TP), total nitrogen (TN), and chlorophyll a (Chla) were measured during all campaigns in all surface sampling sites (Figure 1). Methods for these

analyses are described in detail in Soued et Prairie (2020). Briefly, TP and Chla (extracted with hot ethanol) were analyzed via spectrophotometry, and TN and DOC (filtered at 0.45 μm) were measured on an Alpkem Flow Solution IV autoanalyser and on a Total Organic Carbon analyser 1010-OI respectively.

For each site, we defined the depths of the thermocline and the top and bottom of the metalimnion based on measured temperature profiles using the R package rLakeAnalyzer (Winslow et al., 2018). The epilimnion was defined from the surface

to the top of the metalimnion, and was assumed to be a mixed layer.

## 2.3 Gas concentration, isotopic signature, and water-air fluxes

$CO_2$ and $CH_4$ gas concentrations and isotopic signatures ($\delta^{13}C$) were measured in duplicates at the surface in 36 sites and along vertical profiles in 9 sites (P1 to P9, Figure 1) using the headspace technique described in details in Soued et Prairie (2020). In brief, sampling was done by equilibrating the water sample for two minutes with an air headspace inside a 60 mL syringe. The

gas phase was then injected in a 12 mL pre-vacuumed air-tight vial, and analyzed on a gas chromatograph (Shimadzu GC-8A with a flame ionization detector) for gas concentrations, and on a Cavity Ring Down Spectrometer (CRDS) equipped with a Small Sample Isotopic Module (SSIM, Picarro G2201 -i) for $\delta^{13}CO_2$ and $\delta^{13}CH_4$.

Surface diffusive fluxes of $CO_2$ and $CH_4$ were measured at all surface sampling sites during each campaigns. Flux rates were derived from linear changes in $CO_2$ and $CH_4$ concentrations in a static floating chamber (design described in Soued et Prairie

(2020) and IHA (2010)) connected in a closed loop to a gas analyzer (model UGGA, from Los Gatos Research). Measured gas concentrations, isotopic signature, and fluxes were spatially interpolated to the whole reservoir area by inverse distance weighting (given the absence of a suitable variogram for kriging) using package gstat version 1.1-6 in the R version 3.4.1 software (Pebesma, 2004). Mean values were calculated for each campaign based on the interpolated maps (Soued and Prairie, 2020).



## 2.4 Horizontal GHG inputs

In order to estimate the external horizontal inputs of $CO_2$ and $CH_4$, we considered that the total volume of water inflow and outflow (discharge) were equal, and equivalent to the mean of measured daily discharge (Q, in $m^3$ $d^{-1}$) during each campaign (considering minimal changes in inflow / outflow rates during a campaign). Given that part of the water from the riverine inflows is colder and denser than the reservoir surface layer, only a fraction of the inflowing waters enters the epilimnion of the reservoir, and the rest plunges into the hypolimnion. We estimated the fraction of inflowing water entering the epilimnion ($f_{epi}$) based on temperature profiles at the two ends of the right branch. The areal rate of horizontal $CO_2$ and $CH_4$ inputs (H, in mmol $m^{-2}$ $d^{-1}$) over each section of the reservoir were then calculated following Eq. (1):

$$H = \frac{C_{in} \, Q \, f_{epi}}{A} \tag{1}$$

with A (in $m^2$) the area of the reservoir section considered, and $C_{in}$ (in mmol $m^{-3}$) the concentration of gas in the inflowing water. To estimate gas inputs form the inflows to the branches, $C_{in}$ was considered as the average of gas concentrations measured at the two upstream extremities of the branches (Figure 1). To estimate gas inputs form the branches to the main basin, $C_{in}$ was considered as the gas concentrations measured at the confluence between the two branches (right upstream of the main basin).

## 2.5 Vertical GHG fluxes

We estimated $CO_2$ and $CH_4$ fluxes from the metalimnion to the epilimnion (V) based on the vertical gas diffusivity ($K_z$) and the gradient in gas concentration across the epilimnion-metalimnion interface using Eq. (2) (Wüest and Lorke, 2009):

$$V = K_z \, (C_{meta} - C_{epi}) \tag{2}$$

where $C_{meta}$ and $C_{epi}$ are the gas concentrations at the top of the metalimnion and at the bottom of the epilimnion respectively, measured in profile sites (P1 to P9, Figure 1). $K_z$ was derived from the following Eq. (3) (Osborn, 1980):

$$K_z = \Gamma \frac{\epsilon}{N^2} \tag{3}$$

where $\Gamma$ is the mixing ratio set to 0.2 (Oakey, 1982), $\epsilon$ is the dissipation rate of turbulent kinetic energy, and $N^2$ is the buoyancy frequency. $N^2$ was calculated from measured temperature profiles (YSI probe) using function buoyancy.freq from the rLakeAnalyzer package (Winslow et al., 2018) in the R software (R Core Team, 2017). $\epsilon$ was derived from measured vertical shear microstructure profiles performed in the Aug 2018 campaign in all profile sites shown in Figure 1 (except P1 due to floating logs). Shear profiles were measured with a high frequency (512 Hz) MicroCTD profiler (Rockland Scientific) equipped with two velocity shear probes, two thermistors, tilt and vibration sensors, and a pressure sensor. At each site, the profiler was cast 10 times, 5 with an uprising configuration (from bottom to top of the water column) and 5 with a downward configuration (top to bottom), with a 4 min waiting time between profiles to allow water column disturbance to subside. Data quality check and $\epsilon$ calculation for each profile cast were performed with ODAS v4.3.03 Matlab library (developed by Rockland Scientific) based on Nasmyth shear spectrum (Oakey, 1982), with $\epsilon$ values averaged among the two shear probes



and binned over 1-2 m segments along the profile. For each site, continuous $\epsilon$ profiles were interpolated by fitting a smooth spline through all $\epsilon$ values from replicate casts as a function of depth.

At the epilimnion-metalimnion interface (top of the metalimnion ± 2 m), calculated $\epsilon$ averaged 7.7 x$10^{-9}$ (range from 3.4 x $10^{-9}$ to 1.6 x $10^{-8}$) $m^2$ $s^{-3}$ across all sites sampled with the microCTD, with no significant difference between the main basin and branches sites. In order to estimate vertical gas diffusion, we applied the latter $\epsilon$ average to Eq. (2) and (3) for all measured gas profiles (except P1). The resulting V values for each gas were averaged across sites in the main basin and branches separately to derive estimates of V for each of these two reservoir sections.

**2.6 Sediment GHG inputs**

We calculated $CO_2$ and $CH_4$ inputs from the sediments to epilimnetic waters using gas profiles in sediment cores collected in Apr-May 2017 and Feb-Mar 2018 at 7 sites (P1 to P3 in the reservoir branches and P4, P5, P7, and P9 in the main basin, Figure 1). Sediment cores were collected using a Glew gravity corer attached to a 6 cm wide plastic liner. The liner was pre-drilled with 1 cm holes covered with electric tape at each centimeter up to 40 cm. Upon recovery of the sediment core, 3 mL tip-less syringes were inserted into each hole to extract sediments from each centimeter. The sediment content of each syringe was emptied into a 25 mL glass vial prefilled with 6 mL nano-pure water and immediately air-tight sealed by a butyl rubber stopper crimped with an aluminum cap. Glass vials were pressurized with 40 mL of ambient air using a plastic syringe equipped with a needle to pierce the rubber cap. Glass vials were shaken for 2 min for equilibration before extracting the gas with a syringe and injecting it into a pre-evacuated air-tight vial for analysis of $CO_2$ and $CH_4$ concentrations and isotopic signatures as described above. Additionally, samples of the water overlaying the sediments (~1 cm above) were collected for similar analyses of $CO_2$ and $CH_4$.

Sediment $CO_2$ and $CH_4$ flux rates to the overlaying water column were derived from the vertical gradient of gas concentration measured in the sediment cores and overlaying water. The slope of $CO_2$ or $CH_4$ concentration as a function of depth ($g$, in µmol $L^{-1}$ $m^{-1}$) was calculated for measured values in the first 5 cm of sediments and overlaying water. Most cores exhibited clear linear slopes (p-value < 0.05 and $R^2_{adj}$ > 0.5). In the few cases where a linear slope was not evident, $g$ was replaced by the gradient between the mean gas concentration in the first 3 cm of sediments and the overlaying water. The sediment gas flux rate ($S_f$ in mmol $m^{-2}$ $d^{-1}$) were calculated with Eq. (4):

$$S_f = \frac{g \times d}{p} \tag{4}$$

With $d$ the diffusion coefficient set to 1.5 x $10^{-5}$ $cm^2$ $s^{-1}$ (Donis et al., 2017), and $p$ the sediment porosity assumed to be 2 % based on previous results in Batang Ai (Tan, 2015).

At an ecosystem scale, sediment $CO_2$ and $CH_4$ inputs to the water column (S) were estimated based on average and standard deviation values of sites located in each section of the reservoir (branches and main basin). For each section, mean sediment $CO_2$ and $CH_4$ flux rates were multiplied by the areal ratio of epilimnetic sediments ($A_{epi}$) versus total water area ($A_0$). The latter ratio was calculated based on the hypsometric model (Ferland et al., 2014; Imboden, 1973) as shown in Eq. (5) to (7):





$$q = \left(\frac{z_{max}}{z_{mean}}\right) - 1 \tag{5}$$

$$A_{epi} = A_0 \left(1 - \left(1 - \left(\frac{z_{epi}}{z_{max}}\right)\right)^q\right) \tag{6}$$

$$S = \frac{A_{epi}}{A_0} Sf \tag{7}$$

with q a parameter describing the general shape of the reservoir section, $z_{max}$ and $z_{mean}$ the maximum and mean depths respectively, and $z_{epi}$ the mean depth of the epilimnion (8.0 and 10.5 m in the branches and main basin respectively).

Littoral sediments are known to be a source of $CH_4$ not only through diffusion but also via ebullition. While this emission pathway was found to be important in other reservoirs (Deemer et al., 2016), it is surprisingly low in Batang Ai, equaling less

than 2 % of $CH_4$ surface diffusive emissions, and only 0.1 % of the reservoir total GHG footprint (Soued and Prairie, 2020). Therefore, sediment ebullition was considered negligible in the epilimnetic $CH_4$ budget of Batang Ai.

**2.7 Metabolic rates**

Net metabolic rates of $CO_2$ and $CH_4$ production in the epilimnetic water column were estimated with *in situ* incubations. Incubations were performed in 5 sites (P2 and P3 in the branches and P4, P5 and P7 in the main basin, Figure 1). Water from

3 m deep was pumped into 5 L transparent glass jars with an air tight clamp lid. Before closing, jars were filled from the bottom and allowed to overflow, then sampled for initial $CO_2$ and $CH_4$ concentrations. Closed jars were fixed at 3 m to an anchored line at the sampling site, and incubated in *in situ* temperature and light conditions for 22.0 to 24.2 hours. Upon retrieval, samples of final $CO_2$ and $CH_4$ concentrations were collected from the jars. Volumetric daily rates of net $CO_2$ and $CH_4$ production were calculated based on the difference between final and initial gas concentrations rescaled to a 24 h period.

In addition to incubations, open water high frequency $O_2$ measurements were carried out to derive $CO_2$ metabolism on larger spatial and temporal scales. Rates of GPP, ER, and net ecosystem production (NEP) were estimated in the reservoir surface layer by monitoring and inverse modeling diel $O_2$ changes in the epilimnion. $O_2$ was measured at a one minute interval using high frequency $O_2$ and temperature sensors (model miniDOT from Precision Measurement Engineering). Sensors were deployed in profile sites P1 to P3 in the branches and P4, P5, P7, and P9 in the main basin (Figure 1). Note that not all sites

were sampled in all sampling campaigns. Sensors were attached to an anchored line at a depth between 0.7 and 3 m and deployment time varied between 4 days and two weeks. Upon retrieval of the sensors, a first data quality check and selection was made based on the sensor internal quality index and visual screening. Rates of ecosystem metabolism were then estimated based on an open system diel $O_2$ model (Odum, 1956), where change in $O_2$ concentration is a function of GPP, ER, and air-water gas exchange ($K_{O2}$) following Eq. (8) (Hall and Hotchkiss, 2017):

$$\frac{dO2}{dt} = \frac{GPP}{z_{epi}} + \frac{ER}{z_{epi}} + K_{O2}\left(O_{2_{sat}} - O_2\right) \tag{8}$$

with $O_{2sat}$ the theoretical $O_2$ concentration at saturation considering the *in situ* temperature and atmospheric pressure, and $O_2$ is the actual measured $O_2$ concentration in the water. A detailed description of the model equations can be found in (Hall and



Hotchkiss, 2017). Daily estimates of GPP, ER, and $K_{600}$ (based on $K_{O2}$) were derived by maximum likelihood fitting of the data to the model in Eq. (8) using the R package StreamMetabolizer (Appling et al., 2018). Note that even though the package

used was originally developed for streams, it is easily applicable to lakes / reservoirs systems by replacing the depth of the stream by the depth of the mixed layer. In some cases, where the best predicted $K_{600}$ was negative, the fitting process was rerun with a user defined positive $K_{600}$, either equal to a value estimated for the previous or subsequent day at the same site (range of $0.03 - 0.96$ $d^{-1}$) or fixed to 0.1 $d^{-1}$ (if no other available estimate). When considering the epilimnion depth, predicted values of $K_{600}$ translate into a $1^{st}$ to $3^{rd}$ quantiles range of 1.17 to 5.55 m $d^{-1}$, which is similar to the range of $K_{600}$ values back-calculated

from surface gas flux measurements with the floating chamber technique. A final selection of daily metabolic estimates was done based on the model goodness of fit assessed by calculating Pearson correlation coefficient between modeled and measured $O_2$ values and discarding days  with a correlation lower than 0.9. Based on GPP and ER estimates, we calculated daily NEP as the balance between these two processes, and converted it to net $CO_2$ production rate by assuming an $O_2$:$CO_2$ metabolic quotient of 1.

Areal metabolic rates were derived by integrating volumetric rates over the depth of the epilimnion. Average estimates of areal metabolic rates per campaign were obtained for the branches and main basin by first averaging data within each site and then across sites for each reservoir section. Note that one value derived from incubations was excluded from the calculation of the average net $CH_4$ production rate in the branches due to its high value of initial $CH_4$ concentration (an order of magnitude higher than in all other incubations and all epilimnetic data from this site). The high $CH_4$ concentration, unrepresentative of real

conditions, was probably caused by $CH_4$ contamination during sampling, and triggered a high oxidation rate that would overestimate the real ecosystem average rate if included.

## 3 Results

### 3.1 Physical and chemical properties

Surface water temperature exhibited a marked increase from the inflows to the branches, averaging 27.1 and 30.7 ℃

respectively (Table 1). There was no difference in surface water temperature between the branches and the main basin. The depth of the epilimnion tended to increase and become more stable along the water flow, going from 1.3 (± 1.6) m in Batang Ai river delta, to 8.0 (± 2.3) m in its branch, and 10.6 (± 1.7) m in the main basin (Table 1). Light penetration exhibited the same spatial pattern, with an increasing Secchi depth along the water flow averaging 1.3, 5.1, and 5.5 m in the inflows, branches, and main basin respectively (Table 1). All sections of the study system exhibited oligotrophic water properties (Table

245   1).



## 3.2 Surface GHG concentrations, fluxes, and isotopic signatures

Surface $CO_2$ and $CH_4$ patterns are summarized in Figure 2, presenting campaign averages of spatially interpolated gas concentration, flux, and isotopic signature along the different reservoir sections. Despite the temporal variability, the gas patterns along the water flow are robust, remaining similar throughout time (Figure 2).

Average $CO_2$ air-water flux and surface concentration were systematically higher in the inflows (mean [range]: 135.3 [18.9 – 368.8] mmol m$^{-2}$ d$^{-1}$ and 58.0 [24.5 – 113.0] µmol L$^{-1}$, respectively) compared to the branches (4.7 [-3.4 – 15.2] mmol m$^{-2}$ d$^{-1}$ and 15.4 [12.2 – 19.3] µmol L$^{-1}$) and main basin (7.5 [0.3 – 15.1] mmol m$^{-2}$ d$^{-1}$ and 16.0 [14.2 – 17.7] µmol L$^{-1}$) (Figure 2). Surface $CO_2$ concentration in the reservoir (branches and main basin) was most strongly correlated inversely with water temperature ($R^2_{adj}$ = 0.22, p-value < 0.001, Figure S1 and Table S1). Except for the Apr-Mar 2017 campaign, there was a

modest increase of surface $\delta^{13}CO_2$ towards more enriched values (2.2 to 3.3 ‰) from the inflows to the branches (Figure 2). Similarly, surface $CH_4$ flux and concentration continually decreased along the water channel, being an order of magnitude higher in the inflows compared to the branches, and about twice as high in the branches compared to the main basin (Figure 2). Of all measured water properties, TN was the most strongly linked to reservoir surface $CH_4$ concentration ($R^2$ = 0.14, p-value < 0.001, Figure S1 and Table S1). Surface $\delta^{13}CH_4$ values varied widely, between -83.3 and -47.6 ‰, but did not show a

consistent spatial pattern (Figure 2).

The degree of coupling between $CO_2$ and $CH_4$ followed a clear spatial pattern. While $CO_2$ and $CH_4$ surface concentrations were strongly linked in the inflows ($R^2_{adj}$ = 0.54, p-value = 0.006), they became only weakly correlated in the branches ($R^2_{adj}$ = 0.17, p=0.005) and not correlated at all in the main basin ($R^2_{adj}$ = 0.01, p-value = 0.11) (Figure S2).

## 3.3 Horizontal GHG flow

Horizontal inputs from the inflows to the surface layer of the branches were estimated to vary between 0.34 – 0.71 mol s$^{-1}$ for $CO_2$ and 0.02 – 0.25 mol s$^{-1}$ for $CH_4$. When expressed as rates over the branches surface area (to facilitate comparison with other components), this results in 2.7 – 5.7 and 0.16 – 2.0 mmol m$^{-2}$ d$^{-1}$ for $CO_2$ and $CH_4$ respectively (Table S2 and S3). These values are in the same order of magnitude as surface fluxes calculated in the branches (Figure 3 and Table S2 and S3). However, the effect of horizontal inputs faded spatially, with much lower inputs from the branches to the main reservoir basin, averaging

0.31 and 0.004 mmol m$^{-2}$ d$^{-1}$ for $CO_2$ and $CH_4$, respectively (Figure 3 and Table S2 and S3). For $CH_4$, this fits spatial and temporal surface flux measurements, being systematically higher in the branches, and maximal during the two sampling campaigns with the highest recorded horizontal inputs from the inflows (Table S3). In contrast, $CO_2$ surface flux were typically lower (sometimes negative) in the branches compared to the main basin, despite substantial riverine inputs to the branches (Table S2).





### 3.4 Vertical GHG inputs

Vertical fluxes depend on the gas diffusivity and concentration gradient. Gas diffusivity is a function of the strength of stratification ($N^2$) and energy dissipation rate ($\epsilon$). Measured values of $N^2$ and $\epsilon$ varied widely, from $5.9 \times 10^{-5}$ to $2.3 \times 10^{-3}$ $s^{-2}$ and from $3.4 \times 10^{-9}$ to $1.6 \times 10^{-8}$ $m^2$ $s^{-3}$ respectively, but with no clear differences between the reservoir branches and main basin (Figure S3). Similarly, $CO_2$ and $CH_4$ concentration gradients varied substantially in both space and time (from -18.4 to 94.3 µmol $L^{-1}$ $m^{-1}$ for $CO_2$ and -0.19 to 0.4 µmol $L^{-1}$ $m^{-1}$ for $CH_4$). $CO_2$ concentration generally increased from the epilimnion to the metalimnion as a result of the respiratory $CO_2$ buildup in the deep layer. On rare occasions, an inverse gradient was observed, possibly due to autotrophic activity in the metalimnion. For $CH_4$, metalimnion to epilimnion concentration gradients were generally modest averaging 0.04 µmol $L^{-1}$ $m^{-1}$, and even negative in one third of the profiles leading to the diffusion of epilimnetic $CH_4$ toward deeper layers instead of the reverse. The low to negative $CH_4$ vertical flux results from a highly active methanotrophic layer reducing $CH_4$ concentration in the metalimnion, as evidenced by the strong enrichment effect observed in $\delta^{13}CH_4$ profiles (Figure S4). The combination of vertical diffusivity and gas concentration gradients resulted in vertical fluxes averaging 3.4 (-1.8 to 20.5) mmol $m^{-2}$ $d^{-1}$ for $CO_2$, and 0.01 (-0.01 to 0.09) mmol $m^{-2}$ $d^{-1}$ for $CH_4$, with no significant differences between the reservoir branches and main basin (Figure S3).

### 3.5 GHG inputs from littoral sediments

Areal sediment gas fluxes ranged from 1.2 to 4.0 and -0.29 to 1.10 mmol $m^{-2}$ $d^{-1}$ for $CO_2$ and $CH_4$, respectively (Figure S5), in the range of previously reported values in lakes and reservoirs (Adams, 2005; Algesten et al., 2005; Gruca-Rokosz and Tomaszek, 2015; Huttunen et al., 2006). Sediment fluxes were not different in the branches versus the main basin for both $CO_2$ (mean of 2.2 vs 2.4 mmol $m^{-2}$ $d^{-1}$) and $CH_4$ (mean of 0.17 vs 0.48 mmol $m^{-2}$ $d^{-1}$) (Figure S5). Applying measured averages to the area of epilimnetic sediments in each section yields estimates of sediment inputs to the epilimnion of 0.6 ($\pm$ 0.03) and 0.5 ($\pm$ 0.11) mmol $m^{-2}$ $d^{-1}$ for $CO_2$, and 0.04 ($\pm$ 0.02) and 0.10 ($\pm$ 0.06) mmol $m^{-2}$ $d^{-1}$ for $CH_4$ in the branches and main basin respectively (Figure 3 and Table S2 and S3). These inputs from littoral sediments likely represent an upper limit since they are based on deep pelagic sediment cores (littoral area were too compact for coring), where a higher organic matter accumulation and degradation is expected (Blais and Kalff, 1995; Soued and Prairie, 2020). Even as upper estimates, the calculated rates of sediment GHG inputs remain a relatively modest fraction of the average emissions to the atmosphere for the branches and main basin both for $CO_2$ (13 % and 7 %, respectively) and $CH_4$ (4 % and 23 %, respectively) (Tables S2 and S3).

### 3.6 Metabolism

### 3.6.1 $CO_2$ metabolism

Estimated GPP and ER rates based on diel $O_2$ monitoring ranged from 3.6 to 34.5 µmol $L^{-1}$ $d^{-1}$ and from 5.8 to 29.5 µmol $L^{-1}$ $d^{-1}$ respectively (Figure 4), which is well within the range of reported rates for oligotrophic systems (Bogard and del Giorgio, 2016; Hanson et al., 2003; Solomon et al., 2013). As expected, GPP and ER rates were correlated ($R^2_{adj}$ = 0.23, p-value <





0.001, Figure 4), with photosynthesis stimulating the respiration of produced organic matter. In most cases, GPP exceeded ER, especially in the branches and near aquacultures (Figure 4), where higher nutrients (TP and TN) and Chla concentrations were measured (Table 1).

In the reservoir branches, results from the diel $O_2$ monitoring method suggested systematic net $CO_2$ uptake ranging from -19.2 to -1.4 μmol $L^{-1}$ $d^{-1}$, whereas results from two incubations were slightly above that range (-0.5 to 3.3 μmol $L^{-1}$ $d^{-1}$) (Figure 4). In the main basin, the two methods matched fairly well, with the diel $O_2$ technique capturing a wider variability in net $CO_2$ metabolic rates ranging from -19.2 to 6.1 μmol $L^{-1}$ $d^{-1}$, with an estimated $CO_2$ uptake in 39 out of 54 cases (Figure 4). Areal net $CO_2$ metabolic rates, as the average of the two methods, yielded an ecosystem-scale estimate of -23.2 and -11.8 mmol $m^{-2}$ $d^{-1}$ in the reservoir branches and main basin, respectively (Table S2).

### 3.6.2 CH₄ metabolism

Net metabolic $CH_4$ rates (from incubations) ranged from -0.026 to 0.078 μmol $L^{-1}$ $d^{-1}$, indicating that the $CH_4$ balance in the epilimnion of Batang Ai varied from net oxidation to net production (Table S3). $CH_4$ metabolic rates measured in Batang Ai are within the range of values observed in other systems for oxidation (Guérin and Abril, 2007; Thottathil et al., 2019) and production (Bogard et al., 2014; Donis et al., 2017). No temporal or spatial (branches versus main basin) differences in net metabolic $CH_4$ rate were detected due to a high variability and limited data points.

### 3.7 Ecosystem scale GHG budgets

Estimated sources / sinks of $CO_2$ and $CH_4$ were collated into a budget to evaluate their relative impact on epilimnetic gas concentration and to assess whether their sum matches the measured surface gas fluxes in each section of the reservoir. Figure 3 depicts such reconstruction of the epilimnetic $CO_2$ and $CH_4$ budgets in Batang Ai, as well as the uncertainty limits of each component. While each process varied in time, their relative importance in driving surface fluxes was generally similar from one sampling campaign to another (Table S2 and S3).

### 3.7.1 CO₂ budget

For $CO_2$, epilimnetic sediment inputs had the smallest contribution, being typically an order of magnitude lower than measured surface fluxes in both sections of the reservoir (Figure 3 and Table S2). Vertical $CO_2$ inputs from lower depths on the other hand contributed substantially to surface fluxes in both the branches and the main basin (mean of 0.7 and 4.5 mmol $m^{-2}$ $d^{-1}$ respectively (Figure 3 and Table S2), indicating that hypolimnetic processes impact surface emissions despite the permanent stratification. Horizontal inputs of $CO_2$ were in the same range as vertical inputs (mean of 4.3 mmol $m^{-2}$ $d^{-1}$) in the branches, however, they decreased by an order of magnitude when reaching the main basin (mean of 0.3 mmol $m^{-2}$ $d^{-1}$). Thus, direct $CO_2$ inputs from the inflows notably increase surface flux rates in the reservoir branches but only minimally in the main basin. Net $CO_2$ metabolism was surprisingly variable (switching from negative to positive NEP on a daily time scale), thus making it difficult to derive a sufficiently precise ecosystem-scale estimate to close the epilimnetic budget (Figure 3), despite high



sampling resolution (n = 66 daily metabolic rates). Nevertheless, estimated net water column $CO_2$ metabolism had a considerable impact on the epilimnetic budget, but acts more likely as a $CO_2$ sink rather than a source in Batang Ai (Figure 3 and Table S2). Our best assessment suggests that in the main basin, vertical transport from deeper layers is the main source

sustaining surface $CO_2$ out-flux.

### 3.7.2 CH₄ budget

In contrast with $CO_2$, vertical transport was the smallest source of $CH_4$ to the epilimnion, contributing to less than 2 % to surface fluxes in both reservoir sections (Figure 3 and Table S3). In the branches, sediment inputs and net $CH_4$ metabolic rates were both relatively low (mean of $0.04 \pm 0.02$ and $0.04 \pm 0.05$ mmol m$^{-2}$ d$^{-1}$) and had little impact on the budget, corresponding

each to 4 % of surface fluxes in that section (Figure 3 and Table S3). On the other hand, horizontal inputs were the dominant and most variable source sustaining $CH_4$ emissions in the branches, where the epilimnetic mass balance closed almost perfectly (Figure 3 and Table S3). Despite being the main $CH_4$ source in the branches, horizontal transport was a negligible component in the main basin (< 1 % of the flux, Figure 3 and Table S3). Instead, sediment inputs played a larger role in that section, with a mean of $0.10 (\pm 0.06)$ mmol m$^{-2}$ d$^{-1}$, fueling 22 % of surface emissions in the main basin (Figure 3 and Table S3). As with

$CO_2$, the most variable $CH_4$ component of the mass balance in the main basin was the net metabolism within the epilimnion (mean of $-0.16 \pm 0.19$ mmol m$^{-2}$ d$^{-1}$). Considering all sources, the $CH_4$ budget indicates a deficit of 0.34 mmol m$^{-2}$ d$^{-1}$ to explain measured surface emissions in the main basin (Figure 3 and Table S3).

### 4 Discussion

Our results have highlighted both the importance and the challenges associated with quantifying simultaneously all the

components of the epilimnetic $CO_2$ and $CH_4$ budgets, particularly in a hydrologically complex reservoir system. While mass fluxes (hydrological, sedimentary, and air-water fluxes) are relatively easy to constrain, internal C processing, namely the net metabolic balances between production and consumption of $CO_2$ and $CH_4$ are highly dynamic in both time and space, leading to significant uncertainties when extrapolated to the ecosystem scale. In many studies, some components are only inferred by difference. While convenient from a mass-balance perspective, we argue that assessing all components together is necessary

to clearly identify knowledge gaps as well as sources of uncertainty.

### 4.1 Spatial dynamics of CO₂ and CH₄

The decrease in gas concentration and air-water fluxes along the hydrological continuum observed across sampling campaigns and for both $CH_4$ and $CO_2$ reflects a robust spatial structure of the gases. Concurrently, estimates of the horizontal GHG inputs shows a clear and consistent spatial pattern, being high in the branches but negligible in the main basin. A temporal effect of

riverine inputs was also observed as the two sampling campaigns with the highest horizontal $CH_4$ inputs coincided with the





highest $CH_4$ emissions in the branches (Table S3). All these results concord with the progressively reduced influence of direct GHG catchment inputs and greater preponderance of internal processes along the hydrological flow (Hotchkiss et al., 2015). For $CO_2$, the sharpest change in surface metrics (concentration, flux, and isotopic signature) was observed between the inflows and the reservoir branches (Figure 2). Despite large riverine inputs (Table S2), the branches exhibited low $CO_2$ concentration

and fluxes, as well as an increase in $\delta^{13}CO_2$ matching with high GPP values (Figure 2 and 4). This may reflect increased light availability for phytoplankton when transitioning from the turbid inflows to the reservoir branches (higher Secchi depth, Table 1), a pattern previously reported in other reservoirs (Kimmel and Groeger, 1984; Pacheco et al., 2015; Thornton et al., 1990). While the branch areas are often associated with high $CO_2$ outflux due to riverine inputs (Beaulieu et al., 2016; Paranaíba et al., 2018; Pasche et al., 2019; Roland et al., 2010; Rudorff et al., 2011), they are occasionally observed to have low air-water

flux due to simultaneous nutrient inputs (Paranaíba et al., 2018; Wilkinson et al., 2016). In Batang Ai, inflows have a high nutrients (TP and TN) to DOC ratio compared to the reservoir branches (Table 1), providing higher inputs of nutrients relative to organic matter, and thus likely stimulating primary production more than respiration. This hypothesis is consistent with a higher GPP: ER ratio and mean Chla concentrations measured in the branches compared to the main basin (Figure 4 and Table 1). The variability of $CO_2$ concentration within the reservoir (branches and main basin) was negatively correlated to

temperature, likely due to its effect on GPP (Bogard et al., 2020). This further highlights the important role of primary production in modulating $CO_2$ dynamics throughout the reservoir, and particularly in the branches.

The correlation between surface $CH_4$ surface and TN in the reservoir suggests that primary production may also affect $CH_4$ dynamics. Nutrient content was shown in previous studies to enhance $CH_4$ production in the sediments (Beaulieu et al., 2019; Gebert et al., 2006; Isidorova et al., 2019) and in the oxic water column (Bogard et al., 2014), through its link with algal

production and decomposition. However, $CH_4$ concentration and flux variability were strongly driven by a spatial / hydrological structure, gradually decreasing from the inflows to the main basin. This likely reflects the combined effect of terrestrial inputs and a decreasing contact of water with sediments along the water channel. Surface $\delta^{13}CH_4$ signatures varied substantially but without a consistent spatial pattern (Figure 2), indicating that the surface $CH_4$ pool is shaped by multiple sources / processes (metabolism, riverine, and sediment inputs) varying through space and time.

Overall, the changing relative contribution of sources and processes shaping surface $CO_2$ and $CH_4$ concentrations varies with the system hydro-morphology, from the inflows to the main reservoir basin, and lead to a progressive decoupling between the two gases along the continuum (Figure S2). Thus, while $CO_2$ and $CH_4$ were both influenced by horizontal inputs in the upstream section, their main drivers diverge in the main basin, with $CO_2$ mostly shaped by vertical inputs and aerobic metabolism, and $CH_4$ by sediment inputs and anaerobic metabolism. The spatial patterns reported here highlight the hydrodynamic zonation

common in reservoirs and its diverging effect on $CO_2$ versus $CH_4$ cycling.

### 4.2 $CO_2$ metabolism

Our observation that GPP often exceeded ER (Figure 4) was not unexpected given the very low DOC concentration (< 1 mg $L^{-1}$). Previous work has reported that DOC > 4 mg $L^{-1}$ is required to sustain persistent net heterotrophy and $CO_2$ evasion





(Hanson et al., 2003; Prairie et al., 2002). Throughout the reservoir, we found high day-to-day variability in both ER and GPP, but with no apparent link to weather data (light and rain, data not shown). The absence of such a link at a daily time scale has been previously reported (Coloso et al., 2011), while other studies associated daily variations in metabolism with changes in water inflows carrying nutrients (Pacheco et al., 2015; Staehr and Sand-Jensen, 2007), or thermocline stability regulating hypolimnetic water incursions to the epilimnion. Such variations in thermocline depth are thought to be more common in warm tropical systems (Lewis, 2010), and were observed across sampling campaigns in Batang Ai, especially in the branches where the depth of the mixed layer varied considerably (SD = 2.3 m, Table 1). Hence, hydrological and physical factors may regulate spatial and daily patterns of GPP and ER rates in Batang Ai through their influence on nutrient dynamics.

The accuracy of rates derived from diel $O_2$ monitoring partly depends on the respiratory and photosynthetic quotients (RQ and PQ) assumed for the conversion of metabolic rates from $O_2$ to $CO_2$. A quotient differing from the assumed 1:1 ratio can lead to an under or over-estimation of net $CO_2$ production. The fact that net $CO_2$ metabolic rates were on average higher in incubations, based on direct $CO_2$ measurements compared to diel $O_2$ monitoring (Figure 4 and Table S2), hints at a deviation of the metabolic quotients form unity in Batang Ai. Additionally, surface $O_2$ versus $CO_2$ concentrations shows that the departure of these gases from saturation varies widely around the expected 1:-1 line (Figure 5). $O_2$ oversaturation was observed in 44 % of cases in the main basin and 81 % in the branches (Figure 5), which corresponds with the spatial patterns of net metabolic rates (Figure 4). $CO_2$ oversaturation was also widespread (74 % of cases), making half the surface samples oversaturated in both $O_2$ and $CO_2$. This indicates an excess $O_2$ and / or $CO_2$ that can be due to a PQ and / or a RQ higher than 1, or to external $CO_2$ inputs to the epilimnion (Vachon et al., 2020), for instance from the inflows or the bottom layer (Table S2). Metabolic quotients have been shown to vary widely, depending on the type and magnitude of photochemical and biological reactions at play (Berggren et al., 2012; Lefèvre and Merlivat, 2012; Vachon et al., 2020; Williams and Robertson, 1991). For instance, $CH_4$ oxidation and production, evidently occurring in Batang Ai's epilimnion (Table S2 and S3), diverge from the metabolic $O_2$:$CO_2$ ratio of one, with $CH_4$ oxidation consuming two moles of $O_2$ for each mole of $CO_2$ produced, and acetoclastic methanogenesis producing $CO_2$ without $O_2$ consumption. Even though net $CH_4$ processing rates are a minor portion of the epilimnetic C cycling in Batang Ai (1-2 orders of magnitudes lower than $CO_2$ metabolic rates, Tables S2 and S3), these reactions (and other unmeasured processes) have the potential to alter the $O_2$:$CO_2$ metabolic quotient at an ecosystem scale. The lack of direct measure of metabolic quotients in Batang Ai adds uncertainty to the net $CO_2$ metabolism estimates based on $O_2$ data. The observed decoupling of $O_2$ and $CO_2$ metabolism in Batang Ai highlights the need for a deeper understanding of the biochemical reactions occurring in the epilimnion, and their effect on metabolic quotients.

Overall, our results from Batang Ai reservoir point to water column metabolism as both a major process in the $CO_2$ epilimnetic budget and a challenging one to estimate at an ecosystem scale (Figure 3). Improving this requires a better mechanistic knowledge of the physical and biochemical processes at play and how they interact to shape NEP.





### 4.3 CH$_4$ metabolism

Incubation results exhibited a wide range of net CH$_4$ metabolism: from net oxidation to net production. CH$_4$ oxidation is known to be highly dependent on CH$_4$ availability and is optimal in low oxygen and low light conditions (Borrel et al., 2011; Thottathil et al., 2018, 2019), whereas CH$_4$ production in the oxic water is still poorly understood but have been frequently linked to
phytoplankton growth (Berg et al., 2014; Bogard et al., 2014; Lenhart et al., 2015; Wang et al., 2017). A large variability in results exists among the studies that have assessed the net balance of CH$_4$ metabolism in the water column, with some studies reporting pelagic CH$_4$ production as a largely dominant process (Donis et al., 2017) while others find no trace of it (Bastviken et al., 2008). Based on spatial patterns of surface CH$_4$ concentration and isotopic signature with distance to shore, DelSontro et al. (2018b) showed that, in 30 % of their studied temperate lakes, CH$_4$ oxidation was dominant versus 70 % dominated by
net pelagic production. In Batang Ai, surface $\delta^{13}$CH$_4$ values were highly variable (-82.5 to -47.7 ‰) but mostly uncorrelated with distance to shore, except a positive correlation indicative of oxidation in the Nov-Dec 2016 (R$^2_{adj}$ = 0.29, p-value = 0.01) coinciding with a strong inverse pattern for CH$_4$ concentration (R$^2_{adj}$ = 0.54, p-value < 0.001, Figure 6). This suggests a temporal shift in processes driving surface CH$_4$ patterns. Also, some measured surface $\delta^{13}$CH$_4$ values were lower than the mean $\delta^{13}$CH$_4$ form the sediments (-66.0 ‰, unpublished data), suggesting another highly depleted source of pelagic CH$_4$ in the
system. This is in line with water incubation results often showing positive net CH$_4$ production (Table S3). When reported as mean areal rates, CH$_4$ metabolism ranged from net consumption to net production of CH$_4$ (-0.29 to 0.94 mmol.m$^{-2}$.d$^{-1}$), and had a strong influence on the epilimnetic CH$_4$ budget at the reservoir scale (Figure 3 and Table S3). Results in Batang Ai show that the net balance of CH$_4$ metabolic processes varies widely even within a single system. However, the factors regulating this balance remain largely unknown. Investigating such factors constitute a key step in resolving CH$_4$ budgets in lakes and
reservoirs.

### 4.4 Epilimnetic GHG budgets

For CO$_2$, measured surface fluxes in both reservoir sections fall in the range of possible values estimated by the sum of epilimnetic processes and their uncertainties (Figure 3 and Table S2). However, the averages of those two terms differ substantially, due to negative values of metabolism shifting the mean of the mass balance towards net CO$_2$ consumption
whereas, on average, surface out-flux was measured from the reservoir. This discrepancy indicates either a missing source of CO$_2$ in the budget or the underestimation of one of the processes. While lateral groundwater input is a potential source not explicitly considered, it is probably modest given the small ratio of littoral area to epilimnion volume, and is unlikely to account for the large CO$_2$ deficit in the budget. On the other hand, underestimation of the CO$_2$ metabolic balance is much more likely, given its large variability and uncertainty around its mean value. Additionally, a systematic underestimation of the CO$_2$
metabolic rates derived from the diel O$_2$ method is very possible in Batang Ai given the likely deviation of metabolic quotients around the 1:1 line. As an example, when setting the photosynthetic quotient to 1.2 instead of 1, which remains well within the literature range (Lefèvre and Merlivat, 2012; Williams and Robertson, 1991), the average epilimnetic CO$_2$ mass balance



would increase from -17.7 to 4.3 mmol m$^{-2}$ d$^{-1}$ in the branches and from -6.5 to 6.2 mmol m$^{-2}$ d$^{-1}$ in the main basin, closely matching measured surface fluxes of 4.7 and 7.5 mmol m$^{-2}$ d$^{-1}$ in the respective sections. Thus, constraining the metabolic

component, especially the O$_2$: CO$_2$ quotients, is key for closing the CO$_2$ epilimnetic budget.

In the case of CH$_4$, the epilimnetic mass balance in the branches is surprisingly close to the observed surface flux, largely fueled by horizontal inputs. Hence, CH$_4$ emissions from the branches reflect catchment CH$_4$ loads rather than internal processes. However, in the main basin, these inputs become negligible and the estimated budget does not match measured emissions, indicating a deficit of 0.49 mmolCH$_4$ m$^{-2}$ d$^{-1}$. This amount cannot be explained by a potential underestimation of

horizontal or vertical inputs since they are two orders of magnitude lower. Similarly, sediment inputs would need to be six time higher than estimated to fulfill the budget deficit, which is unlikely given their much lower range of uncertainty. Thus, the most plausible source to close the mass balance in the main basin would be water column CH$_4$ production. Although the estimated CH$_4$ metabolism indicates an average net consumption rather than a net production (-0.16 mmol m$^{-2}$ d$^{-1}$), this mean value is based on only 3 data points and has a high uncertainty associated to it (SE = 0.19 mmol m$^{-2}$ d$^{-1}$, Table S3). Closing

the mass balance would require a net volumetric CH$_4$ production of about 0.03 µmol L$^{-1}$ d$^{-1}$ in the water column of the main basin. This value seems plausible since an equal production rate was measured in one of the incubations, and it is at the low end of the range reported in other systems (Bogard *et al.*, 2014 ; DelSontro *et al.*, 2018b ; Donis *et al.*, 2017). The combination of our results point to water column metabolism as the dominant source of CH$_4$ in the main basin of Batang Ai, sustaining up to 75 % of surface emissions in that reservoir section. However, this process seems highly dynamic and requires more intensive

research into its controls at spatial and temporal scales, commensurate with CH$_4$ emissions.

## 5 Conclusion

The estimated epilimnetic CO$_2$ and CH$_4$ budgets in Batang Ai has helped define the role of different processes in shaping the reservoir surface GHG fluxes to the atmosphere. Results showed that horizontal riverine inputs are important sources of GHG in the reservoir branches (especially for CH$_4$). This creates a coupling between CO$_2$ and CH$_4$ close to the river deltas, which

gradually fades along the water flow, until the surface concentrations of the two gases become completely uncoupled in the main basin being driven by different sources. For instance, vertical inputs from the bottom layer contributed significantly to surface CO$_2$ saturation, while being negligible in the case of CH$_4$ due to metalimnetic oxidation. Inversely, sediment inputs played a notably greater role in sustaining epilimnetic oversaturation of CH$_4$ compared to CO$_2$ in the main basin. Nonetheless, the epilimnetic budgets of both gases were heavily impacted by their respective water column metabolism. This result is likely

representative of large systems with a high volume of water versus sediments, which is common for hydroelectric reservoirs. The metabolic balances of CO$_2$ and CH$_4$ were also extremely variable in space and time, switching from a net production to a net consumption of the gases, and leading to highly uncertain ecosystem-scale estimates. Factors driving these metabolic changes are not well constrained based on current knowledge, highlighting the need for further research on the subject. Overall, this study gives an integrative portrait of the relative contribution of different sources to surface CO$_2$ and CH$_4$ fluxes in a
permanently stratified reservoir including its transition zones (branches). Conclusions and insights derived from this work likely reflect C dynamics in other similar systems, and highlight knowledge gaps guiding future research to better understand and predict aquatic GHG fluxes and regulation.

**Author contribution**

CS contributed to conceptualization, methodology, validation, formal analysis, investigation, data curation, writing - original
draft, writing – review and editing, and project administration. YTP contributed to Methodology, validation, investigation, resources, writing – review and editing, supervision, and funding acquisition.

**Competing interests**

The authors declare that they have no conflict of interest.

**Acknowledgments**

This work was funded by Sarawak Energy Berhad and the Natural Science and Engineering Research Council of Canada (Discovery grant to Y.T.P. and BES-D scholarship to C.S.). We are grateful to Karen Lee Suan Ping and Jenny Choo Cheng Yi for their logistic support and participation in sampling campaigns. We also thank Jessica Fong Fung Yee, Amar Ma'aruf Bin Ismawi, Gerald Tawie Anak Thomas, Hilton Bin John, Paula Reis, Sara Mercier-Blais and Karelle Desrosiers for their help on the field, and Katherine Velghe and Marilyne Robidoux for their assistance during laboratory analyses.

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





**Table 1: Mean (± SD) of physical and chemical variables measured at the surface of the three reservoir sections.**

| Variables | Units | Inflows | Branches | Main basin |
|---|---|---|---|---|
| $z_{epi}$ | m | 1.3 (± 1.6) | 8 (± 2.3) | 10.6 (± 1.7) |
| Secchi | m | 1.2 (± 0.9) | 5.1 (± 1.2) | 5.5 (± 1.2) |
| Temperature | ºC | 27.1 (± 2.5) | 30.7 (± 0.5) | 30.6 (± 0.5) |
| pH | | 6.5 (± 0.3) | 7.2 (± 0.2) | 7.2 (± 0.2) |
| $O_2$ | % | 94.9 (± 7.7) | 102.7 (± 4.5) | 99.3 (± 4.8) |
| DOC | mg L$^{-1}$ | 0.8 (± 0.4) | 0.9 (± 0.2) | 0.9 (± 0.2) |
| TP | µg L$^{-1}$ | 20.7 (± 7.6) | 6.2 (± 1.7) | 5.8 (± 2.6) |
| TN | mg L$^{-1}$ | 0.14 (± 0.04) | 0.12 (± 0.04) | 0.1 (± 0.03) |
| Chla | µg L$^{-1}$ | 2.1 (± 1.7) | 1.7 (± 1) | 1.2 (± 0.5) |

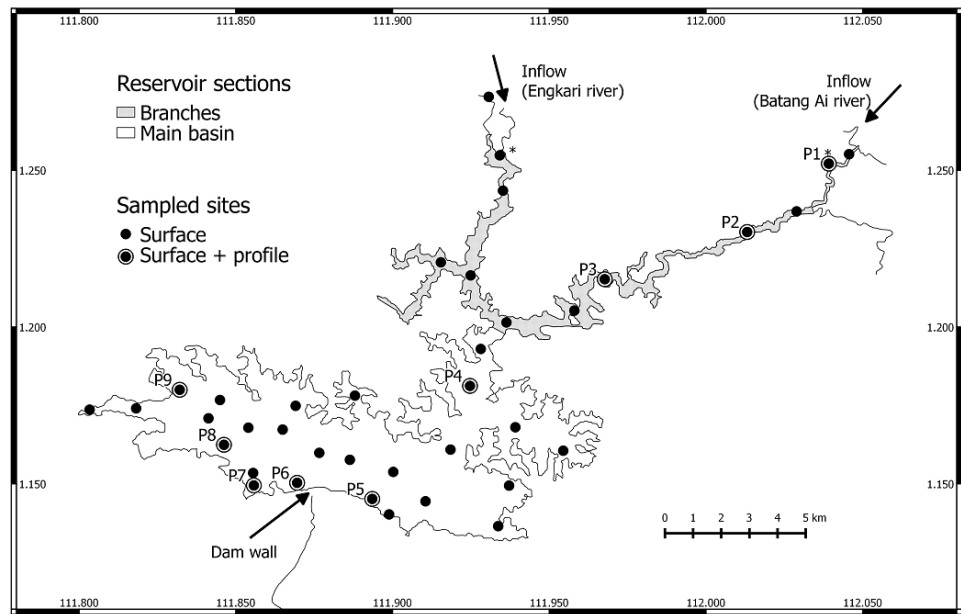


**Figure 1: Map of Batang Ai reservoir with delimited sections (branches and main basin) and sampling points. * Represents sampling points at the branches extremities.**





**Figure 2:** Average of spatially interpolated surface $CO_2$ and $CH_4$ fluxes, concentrations, and isotopic signatures along the
hydrological continuum from the reservoir inflows to the main basin for each sampling campaign.





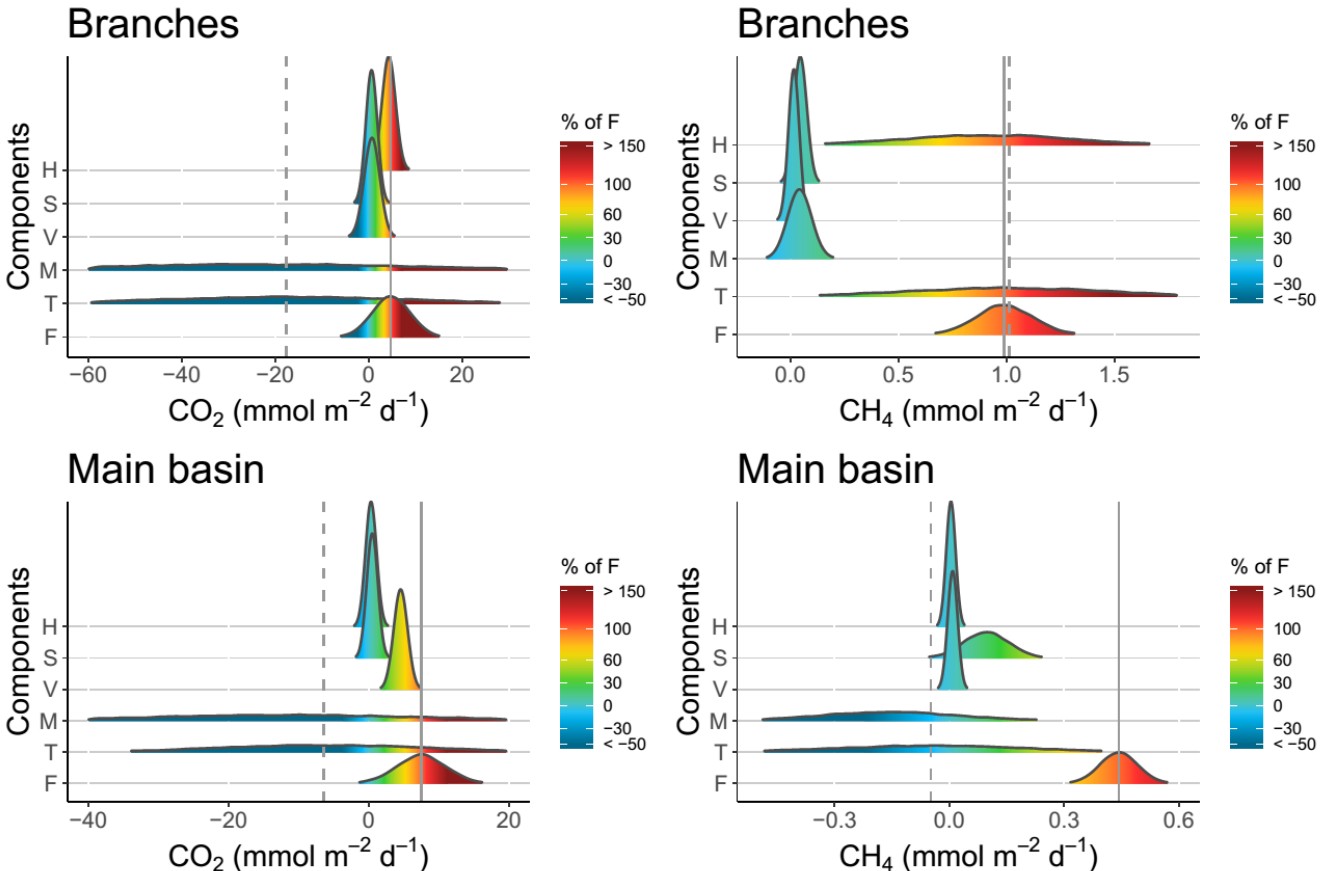

**Figure 3: Density distributions of the different components of CO$_2$ and CH$_4$ surface budget in the reservoir branches and main basin (H = horizontal flow inputs, S = sediment inputs, V = vertical inputs, M = net metabolism, T = sum of all estimated sources and processes in the surface layer, and F = measured surface fluxes). Density curves are based on simulated normal distributions using the mean and standard error of each component. The color scale represents the percentage of the mean surface flux accounted by each component. The solid and dashed grey lines represent the means of F and T respectively.**





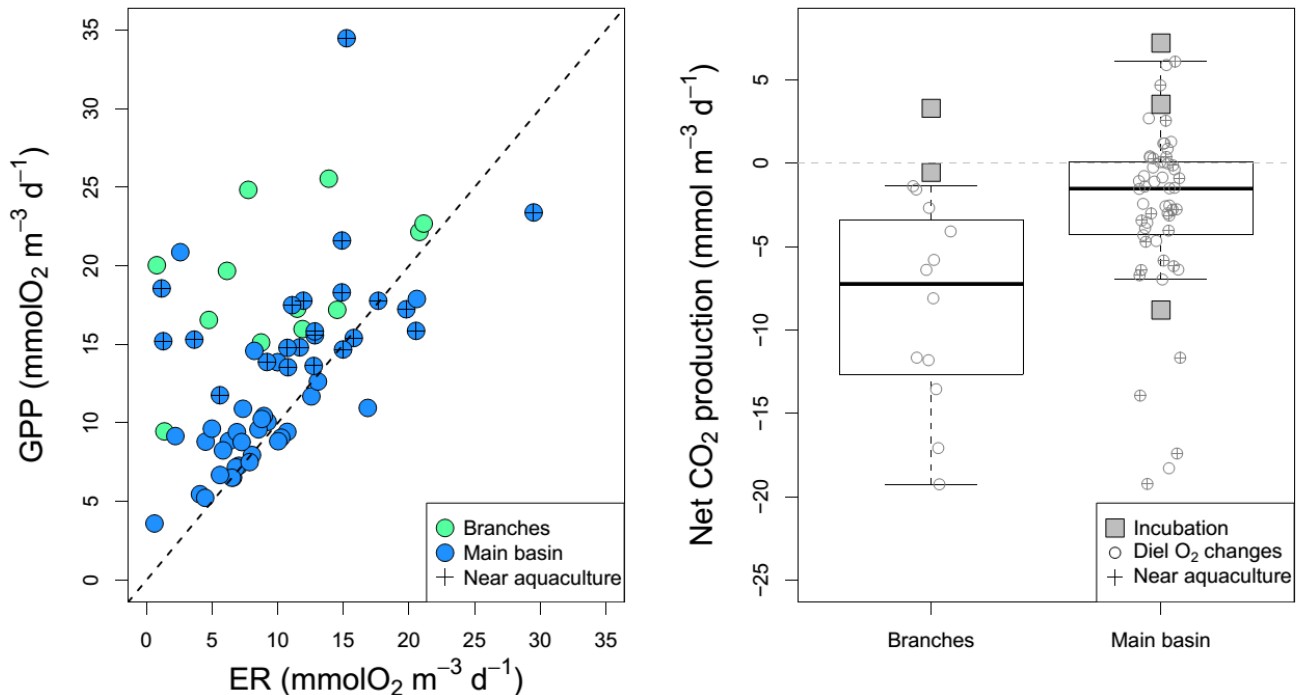

**Figure 4: Epilimnetic daily GPP versus ER rates (left panel) derived from diel $O_2$ changes in the reservoir branches and main basin (including sites near aquacultures), with the 1:1 line (dotted). Boxplots of the corresponding rates of $CO_2$ NEP (right panel) in the branches and main basin, with boxes bounds, whiskers, solid line, and open circles, and squares representing the 25th and 75th percentiles, the 10th and 90th percentiles, the median, single data points (diel $O_2$ method), and incubation derived rates respectively.**



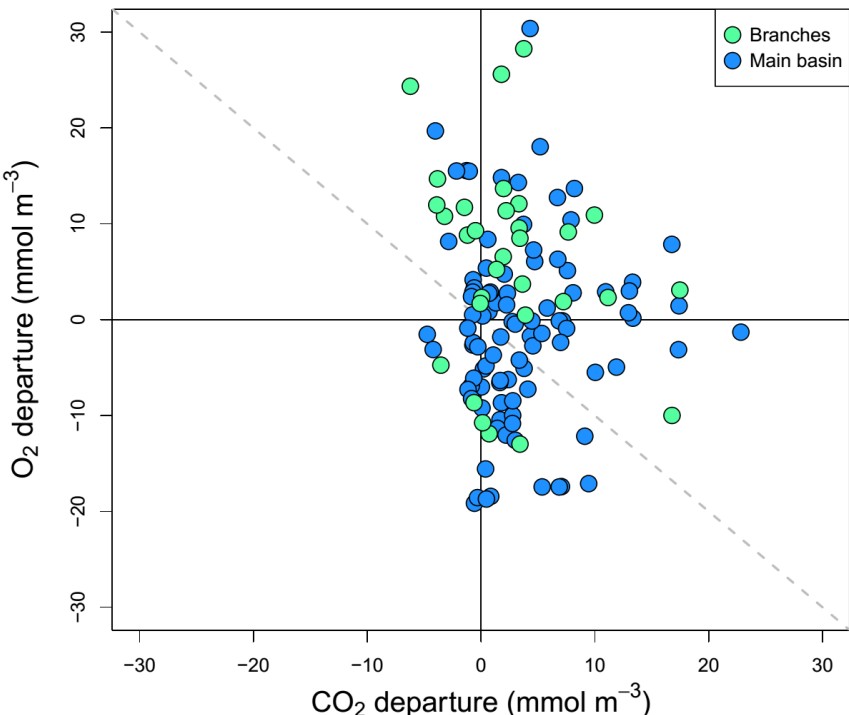

**Figure 5: Surface O₂ versus CO₂ departure from saturation for all sampled surface sites in the reservoir main basin and branches across all sampling campaigns.**

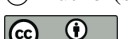



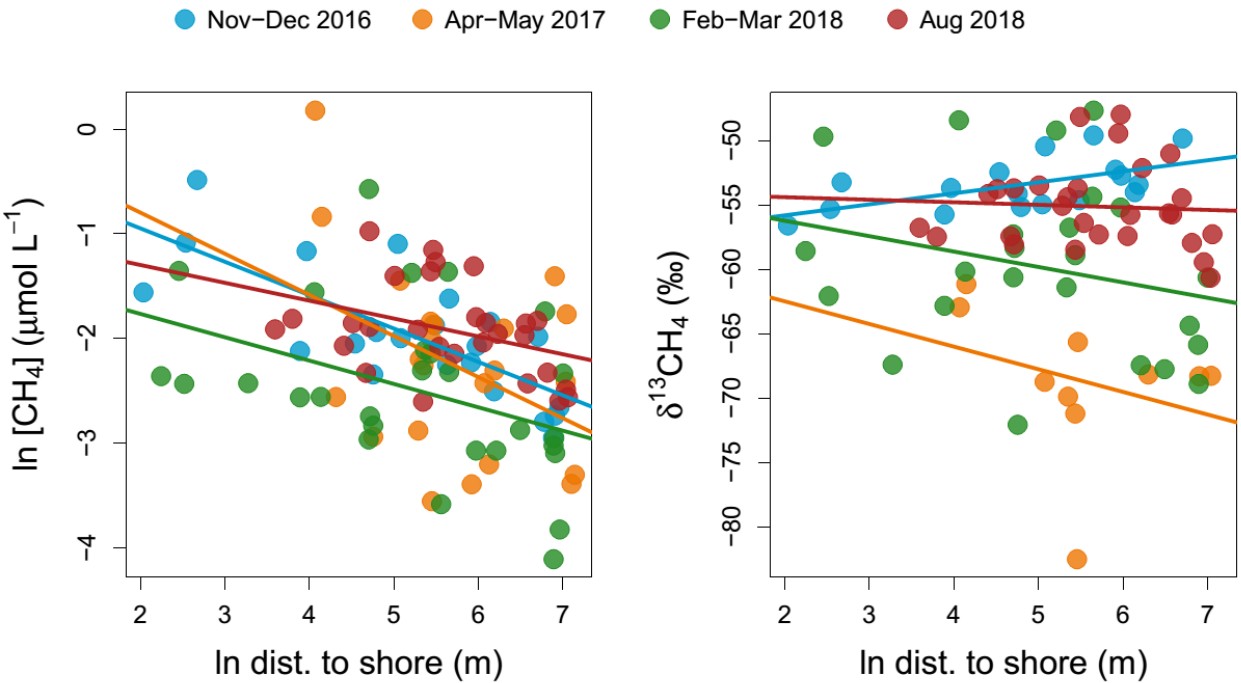

**Figure 6: Regression of CH₄ concentration (left panel) and isotopic signature (right panel) as a function of distance to shore in each sampling campaign in the main reservoir basin. For CH₄ concentration, regressions lines have the following statistics in order of sampling: p-values: < 0.001, 0.06, 0.03, 0.05, and $R^2_{adj}$: 0.54, 0.13, 0.11. For δ¹³CH₄, all regressions had p-values > 0.2 except for the Nov-Dec 2016 campaign with p-value = 0.01 and $R^2_{adj}$ = 0.29.**