# Peer review of "Changing sources and processes sustaining surface CO2 and CH4 fluxes along a tropical river to reservoir system"

_Biogeosciences, 2020_

## Referee Comment (RC1) · Anonymous Referee #1 · 5 Oct 2020

This study is significant in that it attempts to determine the relative importance of various sources of CO2 and CH4 to evasive fluxes from a reservoir. The authors have identified four main sources (lotic inflow, hypolimnion, sediments, water column metabolism) and upscaled measurements and models of these fluxes to determine relative importance to the epilimnion. They find that there is a missing source of CO2 to both the branches and the main basin and a missing source of CH4 to the main basin. I think it's an interesting result that the model can't be closed, but it deserves more attention, perhaps in the title and the abstract. This work falls within the scope of Biogeosciences and generally scientifically sound, with a few exceptions.

[Figure]

My main concern is with the handling of metabolism, and therefore the accuracy of the title and one of the key findings. The authors measure production of CO2 via aerobic metabolism with two methods: bottle incubations and single-station DO measurements. Additionally, the authors measure the metabolic production of CH4 with bottle incubations, which is an interesting and important aspect of this study. However, these metabolism measures are the most uncertain component of their model. The two methods for metabolic CO2 production disagree in sign and by an order of magnitude. And, the results for metabolic CH4 production are highly uncertain with the SE greater than the mean. I think all the authors can conclude is that their estimates for metabolism are highly uncertain and that metabolism has the potential to close the model because the other fluxes are relatively well constrained. Yet the title (Sources and processes sustaining surface CO2 and CH4 fluxes in a tropical reservoir: the importance of water column metabolism) makes it sound like the main finding this that metabolism is the most important source. Additionally, the abstract says, "internal water metabolism remains a dominant driver". I wish that the discussion of this missing sources (like in lines 455-459 of the discussion) was more straightforward in the title, abstract, and results section. Further, Figure 5, which I would interpret as aerobic metabolism not being a dominant control on CO2 dynamics, isn't even mentioned in the results section. I'm also unclear on which CO2 metabolism data is presented in Figure 3.

I'd still like to see a visual more clearly breaking down the relative importance of each source. For example, the abstract says that lotic inflows are responsible for 18%-100% of CO2 and CH4 evasion from the branches. I'm having a hard time making that conclusion from the rest of the paper. For example, the SI table shows that on average 4.3 mmol m-2 d-1 of total CO2 flux from the branches (4.7 mmol m-2 d-) comes from inflows, which would mean that 91% of CO2 evasion is sourced from riverine inputs. If the authors are referring to individual samplings, then influx is between 204% and 18% of CO2 evasion. Fig 3 doesn't clarify things for me either because I find the color gradient confusing. Inflows are colored as 60% to 150+% of CO2 evasion, while

evasion itself is colored as <-50% to >150% of evasive CO2 fluxes.

Minor comments related to scientific content • Should terrestrial inputs (like soil water) be considered another source? Also, because sediment cores couldn't be taken in the littoral zone, it seems like there might be a missing source or two (terrestrial inputs, literal zone) from the model. I do appreciate that the authors discuss that their sediment fluxes might be higher than the average. • Is it justified to assume that water inflow is equal to water outflow of the reservoir? • I don't agree with the statement that the two methods for CO2 metabolism "match fairly well" (line 311). • Line 332 doesn't match the data presented in Table S2. Horizontal inputs are in general an order of magnitude greater than vertical inputs, not in the same range. • The finding presented in lines 399-400 of the discussion section is not presented in the results section. • Lines 411-415 belong in the results section • Figure 6 is not presented in the results section • Figure S2 – linear regression lines shouldn't be drawn if the relationship is insignificant

Additional line-by-line comments Line 7 – the qualifier "two potent GHGs" should be directly after the mention of the gases Line 10 – replace "processes" with "sources" Line 35 – remove ", especially" Line 47 – "associated to highly" should be "associated with highly" Line 49 – I'm not sure that I agree with the idea that GPP and ER are often studied separately. The papers that I read tend to report both. Is there a citation you can use to back up this statement? Line 56 – "lakes" should be "lake" Line 95 – "in 9 sites" should be "at nine sites" Line 105 (and elsewhere) – "Soued et Prairie" should be "Soued and Prairie" or "Soued & Prairie" Line 136 – "inputs form the" should be "inputs from the" Line 166 – "6-cm-wide" liner Line 255 – The placement of the per mille enrichment range is misleading. . . It currently reads as if the range is the $\delta$13CO2 value Line 258 – The R2 value doesn't match the information in the table Line 267 – The values don't match the tables Line 272 – "were" should be "was" Figure S3 – I would expect the legend to introduce the plots in order Line 367 – This citation only applies to CO2 Line 382 – remove "surface"

---

## Referee Comment (RC2) · Anonymous Referee #3 · 24 Nov 2020

This study is novel as it explicitly parses out evasion of CO2 and CH4 (sourced horizontally, vertically, from internal metabolism, and from sediments) from different hydro-morphologic parts of a reservoir and its inflows. The results are interesting as well, showing that the CO2 and CH4 fluxes fundamentally change along a "river to reservoir continuum" and that the overall reservoir budget can not be closed. I think this is all very interesting, however I have some concerns (not dissimilar from Referee #1) about the discussion not sufficiently exploring certain results and the title of the paper being misleading. Overall, this work falls under the scope of Biogeosciences and appears to be scientifically sound.

[Figure]

Perhaps most importantly, the manuscripts' title and abstract arrive at (in my opinion) different conclusions than the conclusion of the manuscript. The manuscript's conclusion does not suggest that the primary finding is 'internal water metabolism remains a dominant driver' as stated in the abstract. Rather, I read the conclusion's primary finding as an "integrative portrait of the relative contribution of different sources to surface $CO_2$ and $CH_4$ fluxes in a permanently stratified reservoir including its transition zones (branches)." I agree with Referee #1 that the massive uncertainties of the metabolism budgets limit the authors' abilities to conclude much from the metabolism values (including its presence in the manuscript's title). I would further argue that this concluding statement is not presented as the main focus of the manuscript's results. The authors allude to the relative contributions of different gas sources in sections 3.3-3.6 but never actually present these relative values. As far as I can tell, they only report the raw fluxes. I suggest the authors focus their statements only on what is directly supported by the results presented in the manuscript, and/or make their presentation of results clearer. I might further add to the results/discussion to sufficiently explore what is being declared in the paper's title, abstract, and conclusion.

Finally, I suggest expanding what is briefly mentioned at line 390-391: relative contributions of sources and processes governing gas concentrations vary with hydromorphology. I think an expanded discussion pertaining to Figure 6, after adding $CO_2$ to the figure (and contextualizing it with Figure 2), would help tremendously here, as the influence of the reservoir hydrodynamics could be explored more thoroughly. Similarly, the authors would benefit from engaging more with the existing literature on spatiotemporal variability in gas concentrations within large lakes/reservoirs (e.g. Chmiel et al 2020; Natchimuthu et al. 2017 as examples).

Following is a list of smaller considerations. Line numbers are in parentheses.

(25) There are many other citations that are relevant here, in addition to DelSontro et al. (2018), which also show inland waters are significant sources of greenhouse gases. I suggest a more thorough reference set. Also, 'surface inland waters' implies you are

also talking about rivers/streams. If so, you need river-specific references as well.

(113 & 119) 'Soued et Prairie' should be 'Soued and Prairie'

(123-124) The reference provided here (another biogeoscience paper by the authors) caused me great confusion because it suggests that the interpolated data analyzed in this manuscript is already published, despite the writing style of the methods suggesting the opposite. This needs to be clarified by the authors because if any of this data/methods are already published, I think that should be explicitly declared in this manuscript.

(Figure 1) I'm not sure I'm convinced that all notable inflow is coming from these two rivers, and this is likely influential when working at the scale of an individual reservoir. I might suggest adding hydrography to Fig 1, or something similar, to show that there aren't really other noteworthy streams/rivers flowing into the reservoir.

(126-128) Along with the previous comment, because you are assuming all inflows are only from these two rivers, reservoir Q could be underestimated (as far as mass balance is concerned). I suggest adding a brief clarifying statement if this is the case.

(136) 'form' should be 'from'

(134) Should specify you are referring to surface area rather than area

(Fig 2) The boxplots are never explained in the main text or caption. Please define these. Also, please include the number of data points composing these boxplots either in the figure or caption.

(Figure 3) There is no explanation of what Figure 3 is actually plotting until section 3.7, after much of the figure's results have been presented. I think this should be mentioned earlier in the manuscript to clarify what is being presented.

(Figure 3) Y-axes need values (i.e. the densities). X-axes need to be scaled uniformly for each gas. In its current form, it is very difficult to compare branch versus main

basin. Similarity, please add subpanel labels and refer to the specific subplot the paper is currently discussing.

(220-221) I'm unfamiliar with this R package but just because you can swap the depth term for the mixed layer depth does not mean that the model is physically realistic for a lotic environment. For example, k600 is often associated with different physical processes in lakes versus rivers and thus modeled differently. This needs an explicit consideration in the manuscript, i.e. why is it ok to run a model built for lentic waters in a lotic environment?

(337-338) Isn't this result just a function of the metabolism uncertainty being so high that it fundamentally effects the aggregate budget ('T' in Figure 3)? Or am I misunderstanding something? This is in line with my earlier comments pertaining to drawing conclusions from these metabolism values.

(367) Do you mean 'hydrological continuum'? Also, it is worth nothing that Hotchkiss et al. (2015), which is cited here, is explicitly a study on the lentic hydrological continuum, and not reservoirs or lakes or any lotic waterbodies. I suggest a more appropo reference.

Chmiel, H. E., Hofmann, H., Sobek, S., Efremova, T., & Pasche, N. (2020). Where does the river end? Drivers of spatiotemporal variability in CO2 concentration and flux in the inflow area of a large boreal lake. Limnology and Oceanography, 65(6), 1161-1174.

Natchimuthu, S., Sundgren, I., Gålfalk, M., Klemedtsson, L., & Bastviken, D. (2017). Spatiotemporal variability of lake pCO2 and CO2 fluxes in a hemiboreal catchment. Journal of Geophysical Research: Biogeosciences, 122(1), 30-49.

―――――――――――――――――――――

---

## Author Comment (AC1) · 7 Dec 2020

**Author's response to comments from anonymous referee #1**

We would like to thank the referee for his/her thorough examination of the manuscript and his/her constructive comments which were used to clarify and improve the manuscript as reported here.

This study is significant in that it attempts to determine the relative importance of various sources of CO2 and CH4 to evasive fluxes from a reservoir. The authors have identified four main sources (lotic inflow, hypolimnion, sediments, water column metabolism) and upscaled measurements and models of these fluxes to determine relative importance to the epilimnion. They find that there is a missing source of CO2 to both the branches and the main basin and a missing source of CH4 to the main basin. I think it's an interesting result that the model can't be closed, but it deserves more attention, perhaps in the title and the abstract. This work falls within the scope of Biogeosciences and generally scientifically sound, with a few exceptions.

We thank the referee for acknowledging the scope and significance of this research.

My main concern is with the handling of metabolism, and therefore the accuracy of the title and one of the key findings. The authors measure production of CO2 via aerobic metabolism with two methods: bottle incubations and single-station DO measurements. Additionally, the authors measure the metabolic production of CH4 with bottle incubations, which is an interesting and important aspect of this study. However, these metabolism measures are the most uncertain component of their model. The two methods for metabolic CO2 production disagree in sign and by an order of magnitude. And, the results for metabolic CH4 production are highly uncertain with the SE greater than the mean. I think all the authors can conclude is that their estimates for metabolism are highly uncertain and that metabolism has the potential to close the model because the other fluxes are relatively well constrained. Yet the title (Sources and processes sustaining surface CO2 and CH4 fluxes in a tropical reservoir: the importance of water column metabolism) makes it sound like the main finding this that metabolism is the most important source. Additionally, the abstract says, "internal water metabolism remains a dominant driver". I wish that the discussion of this missing sources (like in lines 455-459 of the discussion) was more straightforward in the title, abstract, and results section.

This point was raised by the two reviewers, and following their feedback we do agree that the wording used in the interpretation of the results on metabolism and its relative importance were confusing and inadequate in some instances. In this regard, we have made major changes to the new version of the manuscript aiming at better presenting the role of metabolism and shifting the main message to the overall gas budgets. Changes were made throughout the manuscript as follows:

Title: "*Changing sources and processes sustaining surface $CO_2$ and $CH_4$ fluxes along a tropical river to reservoir system*"

Abstract (L.16 - 20): "*Water column metabolism exhibited wide amplitude and range for both gases, making it a highly variable component, but with a large potential to influence surface GHG budgets in either direction. Overall our results show that sources sustaining surface $CO_2$ and $CH_4$ fluxes vary spatially and between the two gases, with internal metabolism acting as a fluctuating but key modulator.*"

Results section 3.7.1 (L.356 - 360): "*Including the metabolism substantially shifts the mean of the $CO_2$ epilimnetic budget (sum of sources and sinks) to a negative value and drastically increases its uncertainty (Fig. 3a, b and Table S2), reflecting a potentially important but poorly resolved role of metabolism in the budget because of its variability. However, given that metabolism acts more likely as a $CO_2$ sink on average, our best assessment suggests that, vertical transport from deeper layers is the main source sustaining surface $CO_2$ out-flux in the main basin of Batang Ai.*"

Discussion section 4.3 (L.475 - 478): "*When reported as mean areal rates, $CH_4$ metabolism ranged from net consumption to net production of $CH_4$ (-0.29 to 0.94 mmol.m$^{-2}$.d$^{-1}$), which reflects its potential in having a high impact, either positive or negative, on the epilimnetic $CH_4$ budget at the reservoir scale (Fig. 3d and Table S3).*"

Discussion section 4.4 (L.507 - 510): "*The combination of our results suggests that water column metabolism could be the dominant source of $CH_4$ in the main basin of Batang Ai, potentially sustaining up to 75 % of surface emissions in that reservoir section.*"

Conclusion section 5 (L.418 - 423): "*Nonetheless, the epilimnetic budgets of both gases presented a high sensitivity to water column metabolism. This result is likely representative of large systems with a high volume of water versus sediments, which is common for hydroelectric reservoirs. However, metabolic balances of $CO_2$ and $CH_4$ were extremely variable in space and time, switching from a net production to a net consumption of the gases, and leading to highly uncertain ecosystem-scale estimates, which emphasizes the key but unconstrained role of metabolism in the overall GHG budgets.*"

Further, Figure 5, which I would interpret as aerobic metabolism not being a dominant control on CO2 dynamics, isn't even mentioned in the results section.

The deviation from 1:1 line in Figure 5 shows indeed that metabolism with a quotient of 1 is not the dominant force controlling $CO_2$ surface concentration which results from several possible factors as explained in the discussion section 4.2. Mentions of metabolism dominance were removed or edited (see previous comment) and Figure 5 was introduced in the result section 3.6.1 (L.330 - 333):

"*To complement metabolic rate data, surface $O_2$ and $CO_2$ departure from saturation were examined in both reservoir sections. $O_2$ oversaturation was observed in 44 % of cases in the main basin and 81 % in the branches (Fig. 5), which corresponds with the spatial patterns of net metabolic rates (Fig. 4b). $CO_2$ oversaturation was also widespread (74 % of cases), making many sampled sites oversaturated in both $O_2$ and $CO_2$ (55 % in the branches and 32 % in the main basin, Fig. 5).*"

I'm also unclear on which CO2 metabolism data is presented in Figure 3.

In the new manuscript version the caption of Figure 3 was edited to clarify the metabolism data presented, and we also added a concise method section 2.8 to clarify the purpose and calculations behind Figure 3 (L.240 - 247):

"***2.8 Epilimnetic GHG budgets***

*Areal rates of horizontal, vertical, sediment, and metabolic inputs were combined into a sum of sources / sinks and compared to the rate of surface gas flux for each gas in each reservoir section. A mean and standard error were calculated for every component of the budgets based on measurements averaged across sites and / or sampling campaigns in order to obtain ecosystem-scale estimates of the components means and uncertainties. In the case of $CO_2$ metabolism, the ecosystem-scale average was calculated as the mean of the two average values derived from the incubation and diel $O_2$ monitoring methods. For every component, density curves were derived considering a normal distribution based on the mean and its standard error in order to visualize the relative magnitude and uncertainty of each ecosystem-scale areal rate (Fig. 3).*"

I'd still like to see a visual more clearly breaking down the relative importance of each source. For example, the abstract says that lotic inflows are responsible for 18%- 100% of CO2 and CH4 evasion from the branches. I'm having a hard time making that conclusion from the rest of the paper. For example, the SI table shows that on average 4.3 mmol m-2 d-1 of total CO2 flux from the branches (4.7 mmol m-2 d-) comes from inflows, which would mean that 91% of CO2 evasion is sourced from riverine inputs. If the authors are referring to individual samplings, then influx is between 204% and 18% of CO2 evasion. Fig 3 doesn't clarify things for me either because I find the color gradient confusing. Inflows are colored as 60% to 150+% of CO2 evasion, while evasion itself is colored as <-50% to >150% of evasive CO2 fluxes.

We agree that based on the previous version of Figure 3, the relative importance of each source was not clearly visually presented and the colored % axis was confusing given its large span (form negative to >100 %). Thus we redesigned Figure 3 which now clearly states the values of mean % contribution from each source. To avoid confusion, we marked <0% for negative areal rates instead of assigning them a negative percent contribution. Also, % contribution are now associated only to the mean of the normal distributions rather than considering the whole uncertainty range, which avoids having large ranges in % and focuses on the average contribution for each component. The range of 18-100% referred to the different sampling campaigns, but we agree that it is confusing and replaced it by the mean (>90%) in the corresponding abstract sentence (L.12 - 14):

"*Results showed that horizontal inputs are an important source of both $CO_2$ and $CH_4$ (> 90 % of surface emissions) in the upstream reservoir branches*."

Minor comments related to scientific content:

- Should terrestrial inputs (like soil water) be considered another source? Also, because sediment cores couldn't be taken in the littoral zone, it seems like there might be a missing source or two (terrestrial inputs, literal zone) from the model. I do appreciate that the authors discuss that their sediment fluxes might be higher than the average. Is it justified to assume that water inflow is equal to water outflow of the reservoir? While soil water inputs (and other lateral flow) were not measured, they were implicitly accounted for as horizontal inputs by considering a steady state where the total amount of inflowing water to the reservoir equals the outflow at the dam. While this assumption is likely a simplification of reality, it seems like the best approach given the limited available data on the hydrography of the system. However, the questions of the referee on this point are very pertinent, thus we edited the method section 2.4 to better explain the choice of this approach, its assumptions, and its limitations (L.124 - 133):
  "*In order to estimate the external horizontal inputs of $CO_2$ and $CH_4$, we considered that the total volume of water inflow and outflow (discharge measured at the dam) were equal, and equivalent to the mean of measured daily discharge (Q, in $m^3$ $d^{-1}$) during each campaign (considering minimal changes in inflow / outflow rates during a campaign). The approach of using discharge as a measure of total water inflow has the advantage of integrating all external flow (rivers, lateral soils, and groundwater) as water inputs to the reservoir. However, the fraction of inflow feeding the reservoir surface versus bottom layer, and its average gas concentration can only be approximated based on measurements from the two main river inlets (Fig. 1) due to the lack of data on other lateral inflows. Given that part of the inflowing water is colder and denser than the reservoir surface layer, only a fraction of it enters the epilimnion of the reservoir branches, and the rest plunges into the hypolimnion. We estimated that fraction ($f_{epi}$) based on temperature profiles in the East river delta and branch (sites P1 and P2, Fig. 1), and assumed it is representative of other water inflows to the reservoir.*"
- I don't agree with the statement that the two methods for CO2 metabolism "match fairly well" (line 311). We agree with the referee that this is an incorrect formulation, we rephrased that sentence as follows (L.325 - 326):
  "*In the main basin, incubation results ranged from -8.8 to 7.2 μmol $L^{-1}$ $d^{-1}$, while the diel O2 technique captured a wider variability in net $CO_2$ metabolic rates from -19.2 to 6.1 μmol $L^{-1}$ $d^{-1}$*"
- Line 332 doesn't match the data presented in Table S2. Horizontal inputs are in general an order of magnitude greater than vertical inputs, not in the same range. The statement was removed.
- The finding presented in lines 399-400 of the discussion section is not presented in the results section. A sentence was added in the result section 3.6.1 (L.322 -323): "*Daily metabolic rates showed no correlation with mean daily rain or light (Kendall rank correlation p-value > 0.1).*" A statement was added in the method section 2.7 for the collection of light data (L.210 - 211): "*...along with light sensors (model HOBO Pendant from Onset).*"
- Lines 411-415 belong in the results section These sentences were edited and moved to the result section 3.6.1 (L.330 - 333):

*"To complement the metabolic rate data, surface $O_2$ and $CO_2$ departure from saturation were examined in both reservoir sections. $O_2$ oversaturation was observed in 44 % of cases in the main basin and 81 % in the branches (Fig. 5), which corresponds with the spatial patterns of net metabolic rates (Fig. 4b). $CO_2$ oversaturation was also widespread (74 % of cases), making many sampled sites oversaturated in both $O_2$ and $CO_2$ (55 % in the branches and 32 % in the main basin, Fig. 5)."*

The discussion section 4.2 was also edited accordingly (L.443 - 445):
*"Additionally, surface $O_2$ versus $CO_2$ concentrations shows that the departure of these gases from saturation varies widely around the expected 1:-1 line, with many surface samples oversaturated in both $O_2$ and $CO_2$, especially in the branches (Fig. 5)."*

- Figure 6 is not presented in the results section
  Description of the results in Figure 6 were added to section 3.2 (L.270 - 274):
  *"In the main basin surface $CH_4$ concentration significantly decreased with distance to shore in Nov-Dec 2016 ($R^2_{adj} = 0.54$, p-value < 0.001), but this correlation was weaker ($R^2_{adj} \leq 0.13$, p-value $\geq$ 0.03) during other sampling campaigns (Fig 6a). Surface $\delta^{13}CH_4$ values varied widely, between -83.3 and -47.6 ‰, but did not show a consistent spatial pattern (Fig. 2f) apart from a positive correlation with distance to shore in the main basin in Nov-Dec 2016 ($R^2_{adj} = 0.29$, p-value = 0.01, Fig. 6b)."*

- Figure S2 – linear regression lines shouldn't be drawn if the relationship is insignificant
  We understand the referee's point here, although from our perspective the regression lines in Figure S2 are not used to represent the significance of the regressions (based on an arbitrary threshold) but rather as a visual representation of the different slopes reflecting the structure of the data spatially. Thus, we argue for keeping the regression lines regardless of significance, but we are willing to reconsider if this is deemed problematic.

Additional line-by-line comments:

- Line 7 – the qualifier "two potent GHGs" should be directly after the mention of the gases
  Fixed (L.7)
- Line 10 – replace "processes" with "sources"
  Fixed (L.9)
- Line 35 – remove ", especially"
  Fixed (L.35)
- Line 47 – "associated to highly" should be "associated with highly"
  Fixed (L.45)
- Line 49 – I'm not sure that I agree with the idea that GPP and ER are often studied separately. The papers that I read tend to report both. Is there a citation you can use to back up this statement?
  The sentence was edited and the statement removed (L.47 - 49).
- Line 56 – "lakes" should be "lake"
  Fixed (L.55)
- Line 95 – "in 9 sites" should be "at nine sites"
  Fixed (L.94)
- Line 105 (and elsewhere) – "Soued et Prairie" should be "Soued and Prairie" or "Soued & Prairie"
  Fixed throughout the manuscript.
- Line 136 – "inputs form the" should be "inputs from the"
  Fixed (L.137)
- Line 166 – "6-cm-wide" liner
  Fixed (L.168)
- Line 255 – The placement of the per mille enrichment range is misleading . . . It currently reads as if the range is the $\delta$13CO2 value
  Fixed (L.266)

- Line 258 – The R2 value doesn't match the information in the table
  The $R^2$ value refers to the linear regression in Figure S1 while Table S1 presents the Kendall correlation coefficient and is cited here for comparison of the link between $CH_4$ and parameters other than TN.
- Line 267 – The values don't match the tables
  One of the value was rounded at the first rather than the second decimal causing the mismatch, this was fixed (L.281).
- Line 272 – "were" should be "was"
  Fixed (L.286)
- Figure S3 – I would expect the legend to introduce the plots in order
  Fixed
- Line 367 – This citation only applies to CO2
  Citations associated to $CH_4$ and to lakes and reservoir systems were added (L.386 - 389):
  "*All these results concord with the a progressively reduced influence of direct GHG catchment inputs and greater preponderance of internal processes along the hydrological flow continuum as observed in river networks (Hotchkiss et al., 2015) and in lakes and reservoirs (Chmiel et al., 2020; Loken et al., 2019; Paranaíba et al., 2018; Pasche et al., 2019).*"
- Line 382 – remove "surface"
  Fixed (L.405)

---

## Author Comment (AC2) · 7 Dec 2020

**Author's response to comments from anonymous referee #3**

We would like to thank referee #3 for taking the time to provide constructive comments, essential to increase the clarity and quality of the manuscript. We have modified the manuscript accordingly as described here.

This study is novel as it explicitly parses out evasion of CO2 and CH4 (sourced horizontally, vertically, from internal metabolism, and from sediments) from different hydromorphologic parts of a reservoir and its inflows. The results are interesting as well, showing that the CO2 and CH4 fluxes fundamentally change along a "river to reservoir continuum" and that the overall reservoir budget cannot be closed. I think this is all very interesting, however I have some concerns (not dissimilar from Referee #1) about the discussion not sufficiently exploring certain results and the title of the paper being misleading. Overall, this work falls under the scope of Biogeosciences and appears to be scientifically sound.

We thank referee #3 for his recognition of the research novelty.

Perhaps most importantly, the manuscripts' title and abstract arrive at (in my opinion) different conclusions than the conclusion of the manuscript. The manuscript's conclusion does not suggest that the primary finding is 'internal water metabolism remains a dominant driver' as stated in the abstract. Rather, I read the conclusion's primary finding as an "integrative portrait of the relative contribution of different sources to surface CO2 and CH4 fluxes in a permanently stratified reservoir including its transition zones (branches)." I agree with Referee #1 that the massive uncertainties of the metabolism budgets limit the authors' abilities to conclude much from the metabolism values (including its presence in the manuscript's title). I would further argue that this concluding statement is not presented as the main focus of the manuscript's results. The authors allude to the relative contributions of different gas sources in sections 3.3-3.6 but never actually present these relative values. As far as I can tell, they only report the raw fluxes. I suggest the authors focus their statements only on what is directly supported by the results presented in the manuscript, and/or make their presentation of results clearer. I might further add to the results/discussion to sufficiently explore what is being declared in the paper's title, abstract, and conclusion.

This point was raised by the two reviewers, and following their feedback we do agree that the wording used in the interpretation of the results on metabolism and its relative importance were confusing and inadequate in some instances. In this regard, we have made major changes to the new version of the manuscript aiming at better presenting the role of metabolism and shifting the main message to the overall gas budgets. Changes were made throughout the manuscript as follows:

Title: "*Changing sources and processes sustaining surface $CO_2$ and $CH_4$ fluxes along a tropical river to reservoir system*"

Abstract (L.16 - 20): "*Water column metabolism exhibited wide amplitude and range for both gases, making it a highly variable component, but with a large potential to influence surface GHG budgets in either direction. Overall our results show that sources sustaining surface $CO_2$ and $CH_4$ fluxes vary spatially and between the two gases, with internal metabolism acting as a fluctuating but key modulator.*"

Results section 3.7.1 (L.356 - 360): "*Including the metabolism substantially shifts the mean of the $CO_2$ epilimnetic budget (sum of sources and sinks) to a negative value and drastically increases its uncertainty (Fig. 3a, b and Table S2), reflecting a potentially important but poorly resolved role of metabolism in the budget because of its variability. However, given that metabolism acts more likely as a $CO_2$ sink on average, our best assessment suggests that, vertical transport from deeper layers is the main source sustaining surface $CO_2$ out-flux in the main basin of Batang Ai.*"

Discussion section 4.3 (L.475 - 478): "*When reported as mean areal rates, $CH_4$ metabolism ranged from net consumption to net production of $CH_4$ (-0.29 to 0.94 mmol.$m^{-2}$.$d^{-1}$), which reflects its potential in having a high impact, either positive or negative, on the epilimnetic $CH_4$ budget at the reservoir scale (Fig. 3d and Table S3).*"

Discussion section 4.4 (L.507 - 510): "*The combination of our results suggests that water column metabolism could be the dominant source of $CH_4$ in the main basin of Batang Ai, potentially sustaining up to 75 % of surface emissions in that reservoir section.*"

Conclusion section 5 (L.418 - 423): "*Nonetheless, the epilimnetic budgets of both gases presented a high sensitivity to water column metabolism. This result is likely representative of large systems with a high volume of water versus sediments, which is common for hydroelectric reservoirs. However, metabolic balances of $CO_2$ and $CH_4$ were extremely variable in space and time, switching from a net production to a net consumption of the gases, and leading to highly uncertain ecosystem-scale estimates, which emphasizes the key but unconstrained role of metabolism in the overall GHG budgets.*"

Finally, I suggest expanding what is briefly mentioned at line 390-391: relative contributions of sources and processes governing gas concentrations vary with hydromorphology. I think an expanded discussion pertaining to Figure 6, after adding CO2 to the figure (and contextualizing it with Figure 2), would help tremendously here, as the influence of the reservoir hydrodynamics could be explored more thoroughly. Similarly, the authors would benefit from engaging more with the existing literature on spatiotemporal variability in gas concentrations within large lakes/reservoirs (e.g. Chmiel et al 2020; Natchimuthu et al. 2017 as examples).

We agree with the reviewer on expanding the literature on spatiotemporal variability, and thus included additional references in the text section 4.7 (Chmiel et al., 2020; Loken et al., 2019; Lupon et al., 2019; Natchimuthu et al., 2017; Paranaíba et al., 2018; Rasilo et al., 2017). We also expanded the discussion on the relative contribution of sources along the hydrological continuum by editing the last paragraph of section 4.1 (L.413 - 427):

"*The changing relative contribution of sources and processes shaping surface $CO_2$ and $CH_4$ concentrations varies with the system hydro-morphology, from the inflows to the main reservoir basin, and lead to a progressive decoupling between the two gases along the continuum (Fig. S2). The observed $CO_2$ and $CH_4$ coupling in the inflows and branches is associated to a common catchment source, as previously reported in other systems including soil-water (Lupon et al., 2019), streams (Rasilo et al., 2017), and lake and reservoir inflow areas (Loken et al., 2019; Paranaíba et al., 2018). Indeed, horizontal inputs are the main source of both $CO_2$ and $CH_4$ in the upstream reaches of Batang Ai, accounting on average for 91 and 92 % of their respective surface out-flux in the branch section (Fig. 3a, c and Tables S2 and S3). However, when reaching the main basin, driving sources diverge between the two gases, with vertical inputs from the bottom layer supporting on average 60 % of $CO_2$ compared to 2 % of $CH_4$ fluxes, while sediment inputs sustained 7 versus 23 % of $CO_2$ and $CH_4$ fluxes respectively in that section. This decoupling partly results from the two gases having distinct metabolic pathways: mainly aerobic for $CO_2$ and anaerobic for $CH_4$, leading to their sources and sinks being spatially disconnected in the main basin. Consequently, sediments being a mostly anaerobic environment are a more important source of $CH_4$ relative to $CO_2$, while the metalimnetic layer being oxic-hypoxic acts as a sink of $CH_4$ and source of $CO_2$ via aerobic $CH_4$ oxidation (Fig. S4). Overall, the spatial patterns reported here highlight the hydrodynamic zonation common in reservoirs and its diverging effect on $CO_2$ versus $CH_4$ cycling.*"

Concerning Figure 6, we would like to clarify that it represents $CH_4$ patterns in the main basin only, so it is not meant to explore the effect of hydrodynamic changes throughout the river to reservoir continuum. The aim of this figure is rather to explore evidences of $CH_4$ production in the lateral sediment versus the water column using distance to shore as a proxy for distance from lateral sediment as a potential $CH_4$ source. Since lateral sediment are not known as a large source of $CO_2$ we don't feel that including $CO_2$ in Figure 6 is appropriate, unless we have misunderstood the idea behind the suggestion of the referee here.

Following is a list of smaller considerations. Line numbers are in parentheses.

- (25) There are many other citations that are relevant here, in addition to DelSontro et al. (2018), which also show inland waters are significant sources of greenhouse gases. I suggest a more thorough reference set.

Also, 'surface inland waters' implies you are also talking about rivers/streams. If so, you need river-specific references as well.

Two references were included here (Bastviken et al., 2011; Raymond et al., 2013), including river systems (L.25).

- (113 & 119) 'Soued et Prairie' should be 'Soued and Prairie'

Fixed throughout the manusript

- (123-124) The reference provided here (another biogeoscience paper by the authors) caused me great confusion because it suggests that the interpolated data analyzed in this manuscript is already published, despite the writing style of the methods suggesting the opposite. This needs to be clarified by the authors because if any of this data/methods are already published, I think that should be explicitly declared in this manuscript.

We apologize for the confusion. The data on surface gas fluxes is already published and reused in a different context here. This was clarified in the method section 2.3 by adding the sentence (L.115 - 116):

"*Surface gas flux data used in this study are described Surface gas flux data used in this study are described in more details in Soued and Prairie (2020), a previous study on the C footprint of Batang Ai reservoir.*"

- (Figure 1) I'm not sure I'm convinced that all notable inflow is coming from these two rivers, and this is likely influential when working at the scale of an individual reservoir. I might suggest adding hydrography to Fig 1, or something similar, to show that there aren't really other noteworthy streams/rivers flowing into the reservoir. (126-128) Along with the previous comment, because you are assuming all inflows are only from these two rivers, reservoir Q could be underestimated (as far as mass balance is concerned). I suggest adding a brief clarifying statement if this is the case.

We agree with the referee that the two rivers represent a large part of the inflowing water but most likely not the entire mass of inflowing water. Unfortunately there is no available information on other potential inflows like smaller rivers or groundwater. However, to counter this problem when calculating horizontal inputs, we used total discharge (measured at the dam outflow) as a representative measure of total inflow to the reservoir (rather than using flow measurements from the two rivers). To clarify this, we edited the method section 2.4 as follows (L.124 - 133):

"*In order to estimate the external horizontal inputs of $CO_2$ and $CH_4$, we considered that the total volume of water inflow and outflow (discharge measured at the dam) were equal, and equivalent to the mean of measured daily discharge (Q, in $m^3$ $d^{-1}$) during each campaign (considering minimal changes in inflow / outflow rates during a campaign). The approach of using discharge as a measure of total water inflow has the advantage of integrating all external flow (rivers, lateral soils, and groundwater) as water inputs to the reservoir. However, the fraction of inflow feeding the reservoir surface versus bottom layer, and its average gas concentration can only be approximated based on measurements from the two main river inlets (Fig. 1) due to the lack of data on other lateral inflows. Given that part of the inflowing water is colder and denser than the reservoir surface layer, only a fraction of it enters the epilimnion of the reservoir branches, and the rest plunges into the hypolimnion. We estimated that fraction ($f_{epi}$) based on temperature profiles in the East river delta and branch (sites P1 and P2, Fig. 1), and assumed it is representative of other water inflows to the reservoir.*"

- (136) 'for m' should be 'from'

Fixed (L.137)

- (134) Should specify you are referring to surface area rather than area

Fixed (L.136)

- (Fig 2) The boxplots are never explained in the main text or caption. Please define these. Also, please include the number of data points composing these boxplots either in the figure or caption.

The boxplot were actually formed by the points in the plots (each boxplot was composed by the 4 points representing each sampling campaigns), so since they were showing redundant information already represented by the points we decided to remove them.

- (Figure 3) There is no explanation of what Figure 3 is actually plotting until section 3.7, after much of the figure's results have been presented. I think this should be mentioned earlier in the manuscript to clarify what is being presented.

In the new manuscript version we added a concise method section 2.8 to clarify the purpose and calculations behind Figure 3 (L.240 - 247):

"*2.8 Epilimnetic GHG budgets*

*Areal rates of horizontal, vertical, sediment, and metabolic inputs were combined into a sum of sources / sinks and compared to the rate of surface gas flux for each gas in each reservoir section. A mean and standard error were calculated for every component of the budgets based on measurements averaged across sites and / or sampling campaigns in order to obtain ecosystem-scale estimates of the components means and uncertainties. In the case of $CO_2$ metabolism, the ecosystem-scale average was calculated as the mean of the two average values derived from the incubation and diel $O_2$ monitoring methods. For every component, density curves were derived considering a normal distribution based on the mean and its standard error in order to visualize the relative magnitude and uncertainty of each ecosystem-scale areal rate (Fig. 3).*"

- (Figure 3) Y-axes need values (i.e. the densities). X-axes need to be scaled uniformly for each gas. In its current form, it is very difficult to compare branch versus main basin. Similarity, please add subpanel labels and refer to the specific subplot the paper is currently discussing.
  Figure 3 was redesigned and X-axes scaled uniformly as suggested. Also, subpanel identification were added in all figures and referred to in the text. Concerning the Y-axis of Figure 3, it is not possible to add a common axis of densities since each normal distribution is on a separate horizontal axis. Though we believe showing densities might not be essential in this case since it would not offer substantial additional information needed to convey the message the Figure presents.

- (220-221) I'm unfamiliar with this R package but just because you can swap the depth term for the mixed layer depth does not mean that the model is physically realistic for a lotic environment. For example, k600 is often associated with different physical processes in lakes versus rivers and thus modeled differently. This needs an explicit consideration in the manuscript, i.e. why is it ok to run a model built for lentic waters in a lotic environment?
  We understand the referee's questioning on this matter thus we added clarifying statement in the method section 2.7 (L.221 - 225):
  "*Note that even though the package used was originally developed for streams, it is easily transferable to lakes given that the model used (Eq. (8)) is generalized for all water bodies, with the parameter $z_{epi}$ describing the depth of a mixed water column of either a lentic or lotic system, and with the K600 estimate relying only on data fitting to the model and not on system type.*"
  As mentioned, $K_{600}$ estimates in the model are derived from maximum likelihood fitting of the data to the model rather being modeled by additional variables related to weather or hydrology, making the model independent of system type.

- (337-338) Isn't this result just a function of the metabolism uncertainty being so high that it fundamentally effects the aggregate budget ('T' in Figure 3)? Or am I misunderstanding something? This is in line with my earlier comments pertaining to drawing conclusions from these metabolism values.
  We agree with the referee that this statement is misleading and changed it to highlight the potential influence of metabolism rather than its definite role in the budget (L.356 - 360):
  "*Including the metabolism substantially shifts the mean of the $CO_2$ epilimnetic budget (sum of sources and sinks) to a negative value and drastically increases its uncertainty (Fig. 3a, b and Table S2), reflecting a potentially important but unresolved role of metabolism in the budget.*"

- (367) Do you mean 'hydrological continuum'? Also, it is worth nothing that Hotchkiss et al. (2015), which is cited here, is explicitly a study on the lentic hydrological continuum, and not reservoirs or lakes or any lotic waterbodies. I suggest a more appropo reference.
  This was changed to hydrological continuum which is in fact a more appropriate word. We also added references associated to lotic systems while keeping the original reference deemed pertinent here since it addresses more explicitly the contribution of external versus internal $CO_2$ to surface flux. The sentence was edited as follows (L.386 - 389):
  "*All these results concord with the a progressively reduced influence of direct GHG catchment inputs and greater preponderance of internal processes along the hydrological flow continuum as observed in river*

*networks (Hotchkiss et al., 2015) and in lakes and reservoirs (Chmiel et al. 2020; Loken et al., 2019; Paranaíba et al., 2018; Pasche et al., 2019).*"

---

## Referee Report (RR1)

I thank the authors for a thorough and thoughtful response to my comments. The authors largely addressed my comments with responses that I support (particularly shifting the paper's focus to the overall gas budgets). I have two additional comments (mostly minor) left to address, still regarding the treatment of 1) the metabolism results and 2) a discussion on reservoir morphometry. Line numbers refer to the most recent version of the manuscript.

1) Thank you for clarifying and amending the text and title to reflect the uncertainty in the net metabolism results (namely Figure 3 and discussion pertaining to that). The % relative contributions added to Figure 3 are very helpful. However, I still struggle with the '% of mean total flux' for M and T (i.e. the percents in Figure 3 for net metabolism and the sum of estimated sources). Given the uncertainty of metabolism (often greater than its mean in Table S2), I still disagree with presenting only results that include the mean of these density curves. I suggest presenting two values for M (and thus T): one including net metabolism and one that calculates metabolism by closing the mass-balance. Put another way, I would like to see the 75% value at line 509 presented for all Figure 3 subpanels and subsequently discussed in the manuscript in addition to the discussion at 501-510. I also encourage a brief discussion on how this high uncertainty implicates your argument at line 378-380: "In many studies, some components are only inferred by difference. While convenient from a mass-balance perspective, we argue that assessing all components together is necessary to clearly identify knowledge gaps as well as sources of uncertainty." I agree with this statement in principal but the authors then never explicitly discuss how the very high uncertainties in their metabolism values force them to also present mass-balance results (i.e. line 509). I think you do a good job of noting this discrepancy, but you should add a few sentences explicitly engaging with this argument considering your results.

2) I think your expansion on the spatiotemporal variability in gas concentrations at line 413 is great, however it is still not explicitly addressing the hydromorphology/morphometry of the reservoir. I think a study so squarely focused on relative changes from reservoir inflows to the main basin needs to comment on changes of reservoir morphometry, i.e. possible implications of changing reservoir volume, depth, shape, distance from horizontal inputs, etc. I understand you do not have the data to robustly analyze this, but some sort of literature-informed speculation is suggested. I think an expanded Figure 6 (as a separate Figure) could help parse out some influences here, though I understand that is not the focus of the Figure and do not think it is necessary (thank you for the clarification).

---

## Author Response (AR2)

**Author's response to comments from anonymous referee #1**

I think the authors did a good job of responding to most the reviewer comments. I especially appreciate that the authors reframed the title and abstract to match the key results.

We are grateful for the referee's feedback which pushed us to substantially improve the manuscript.

One remaining concern is Figure 3. Some questions that are still coming up for me, as a reader, include:

-Why is a flux accounting for 100% surface flux the same tan color as fluxes that account for <10% of this metric? Maybe the color scale should be removed?

In light of this comment we changed the color scale to represent the direction of the rate (sink versus source) rather than the rate value, which should be visually more useful to the reader.

-In the figure legend, what does "The x-axes and color scales represent the areal rate of CO2/ CH4" mean?

This part of the figure legend was changed to avoid confusion:

"*The x-axes represent the areal rate of $CO_2$ or $CH_4$ and the color scale indicates the sign of the rate*"

This is not a deal breaker but perhaps still a suggestion that authors could use: In general, I find the discussion section well written and, as a reader, would appreciate the other sections to be written in this clear and concise style.

We did our best to make all sections of the manuscript clear and concise. We modified slightly the text throughout to improve the flow.

Other comments:

Line 138: form --> from

Fixed

Line 270: add comma after basin

Done

-The supplemental materials should be organized in order of reference in the text

This was fixed, Tables S2 and S3 were moved up right after Figure S2 to respect their order of appearance in the text.

-Some of the plot labels have typos (e.g., Figure 4 mmolO2, Figure S5 mmol.m-2.d-1).

We have now made the units labelling uniform.

-I also found the plot labels of CO2/CH4 confusing, initially interpreting it as a ratio.

To avoid confusion, "$CO_2/CH_4$" was replace by "$CO_2$ or $CH_4$" in the caption of Figure 3 and axis label of Figure S4.

Line 402: it seems like a negative relationship between [CO2] and temp could be due to solubility?

The referee raises a good point here, although, when accounting for solubility, the [CO2] – temperature relationship is still strongly significant (p-value $< 0.001$ and $R^2_{adj} = 0.19$).

Line 434-5: Are there any citations to back-up the statement that other studies have attributed variable metabolism rates to "thermocline stability regulating hypolimnetic water incursions to the epilimnion"?

A citation was added here (Coloso et al., 2011) Line 438

**Author's response to comments from anonymous referee #3**

I thank the authors for a thorough and thoughtful response to my comments. The authors largely addressed my comments with responses that I support (particularly shifting the paper's focus to the overall gas budgets).

We appreciate the referee's acknowledgement of the work done and thank her / him for the pertinent and useful feedback.

I have two additional comments (mostly minor) left to address, still regarding the treatment of 1) the metabolism results and 2) a discussion on reservoir morphometry. Line numbers refer to the most recent version of the manuscript.

1) Thank you for clarifying and amending the text and title to reflect the uncertainty in the net metabolism results (namely Figure 3 and discussion pertaining to that). The % relative contributions added to Figure 3 are very helpful. However, I still struggle with the '% of mean total flux' for M and T (i.e. the percents in Figure 3 for net metabolism and the sum of estimated sources). Given the uncertainty of metabolism (often greater than its mean in Table S2), I still disagree with presenting only results that include the mean of these density curves. I suggest presenting two values for M (and thus T): one including net metabolism and one that calculates metabolism by closing the mass-balance. Put another way, I would like to see the 75% value at line 509 presented for all Figure 3 subpanels and subsequently discussed in the manuscript in addition to the discussion at 501-510. I also encourage a brief discussion on how this high uncertainty implicates your argument at line 378-380: "In many studies, some components are only inferred by difference. While convenient from a mass-balance perspective, we argue that assessing all components together is necessary to clearly identify knowledge gaps as well as sources of uncertainty." I agree with this statement in principal but the authors then never explicitly discuss how the very high uncertainties in their metabolism values force them to also present mass-balance results (i.e. line 509). I think you do a good job of noting this discrepancy, but you should add a few sentences explicitly engaging with this argument considering your results.

Following the suggestions of the referee, we modified Figure 3 to present the contribution of metabolism derived from a mass balance approach in addition to the empirical estimates. We also added several sentences in the discussion section 4.4 discussing the two approaches and how they complement each other:

*"Another way to decipher the role of metabolism, given its high uncertainty, is by difference in a mass balance exercise. Assuming mean estimates of all other components are accurate, $CO_2$ net metabolic rates would have to be equal to -0.8 and 2.1 mmol $m^{-2}$ $d^{-1}$ in the branches and main basin respectively for the mass balance to close. This corresponds to a contribution of -18 and 28*

*% to the surface $CO_2$ flux in the respective sections (Fig. 3a, b), suggesting a substantial impact of metabolism on the $CO_2$ epilimnetic budget.*" (Lines 499 - 502)

"*This mass balance approach suggests that water column metabolism could be the dominant source of $CH_4$ in the main basin of Batang Ai, potentially sustaining up to 75 % of surface emissions in that reservoir section (Fig. 3d). Even though this deductive approach is an indirect assessment of water column $CH_4$ metabolism, it emphasizes its likely key role in the reservoir epilimnetic $CH_4$ budget, while measured metabolic rates highlight the wide variability of this process and the need for more intensive research into its controls at spatial and temporal scales.*

*The combination of empirical and mass balance approaches in this study provide not only a partitioning of the contribution of each source / sink in sustaining surface $CO_2$ and $CH_4$ fluxes, but also a clear picture of the uncertainties and challenges associated to the estimation of each component.*" (Lines 514 - 522)

2) I think your expansion on the spatiotemporal variability in gas concentrations at line 413 is great, however it is still not explicitly addressing the hydromorphology/morphometry of the reservoir. I think a study so squarely focused on relative changes from reservoir inflows to the main basin needs to comment on changes of reservoir morphometry, i.e. possible implications of changing reservoir volume, depth, shape, distance from horizontal inputs, etc. I understand you do not have the data to robustly analyze this, but some sort of literature-informed speculation is suggested. I think an expanded Figure 6 (as a separate Figure) could help parse out some influences here, though I understand that is not the focus of the Figure and do not think it is necessary (thank you for the clarification).

We elaborated the discussion on the impact of the main basin versus branches morphometries on horizontal inputs and other $CO_2$ and $CH_4$ sources:

"*The hydro-morphometry of these channels can explain the large impact of horizontal inputs in the branch section, which is characterized by a relatively small ratio of water to catchment area and a direct connection to the major river inflows creating a strong link between the catchment and the branches. However, when reaching the main basin, this link weakens due to a longer distance from river inflows and the dilution of horizontal inputs in a larger water volume. Thus, in the main basin, $CO_2$ and $CH_4$ are mostly driven by internal sources, diverging between the two gases, with vertical inputs from the bottom layer supporting on average 60 % of $CO_2$ compared to 2 % of $CH_4$ fluxes, while sediment inputs sustained 7 versus 23 % of $CO_2$ and $CH_4$ fluxes respectively in that section.*" (Lines 418 - 4265)